# Wavelength-multiplexed hook nanoantennas for machine learning enabled mid-infrared spectroscopy

Zhihao Ren [1,2,3,5], Zixuan Zhang[1,2,3,5], Jingxuan Wei [1,2,3], Bowei Dong [1,2,3] & Chengkuo Lee [1,2,3,4✉]

Infrared (IR) plasmonic nanoantennas (PNAs) are powerful tools to identify molecules by the IR fingerprint absorption from plasmon-molecules interaction. However, the sensitivity and bandwidth of PNAs are limited by the small overlap between molecules and sensing hotspots and the sharp plasmonic resonance peaks. In addition to intuitive methods like enhancement of electric field of PNAs and enrichment of molecules on PNAs surfaces, we propose a loss engineering method to optimize damping rate by reducing radiative loss using hook nanoantennas (HNAs). Furthermore, with the spectral multiplexing of the HNAs from gradient dimension, the wavelength-multiplexed HNAs (WMHNAs) serve as ultrasensitive vibrational probes in a continuous ultra-broadband region (wavelengths from 6 µm to 9 µm). Leveraging the multi-dimensional features captured by WMHNA, we develop a machine learning method to extract complementary physical and chemical information from molecules. The proof-of-concept demonstration of molecular recognition from mixed alcohols (methanol, ethanol, and isopropanol) shows 100% identification accuracy from the micro-fluidic integrated WMHNAs. Our work brings another degree of freedom to optimize PNAs towards small-volume, real-time, label-free molecular recognition from various species in low concentrations for chemical and biological diagnostics.

[1] Department of Electrical and Computer Engineering, National University of Singapore, Singapore 117583, Singapore. [2] Center for Intelligent Sensors and MEMS (CISM), National University of Singapore, Singapore 117583, Singapore. [3] NUS Suzhou Research Institute (NUSRI), Suzhou, Jiangsu 215123, China. [4] NUS Graduate School - Integrative Sciences and Engineering Programme (ISEP), National University of Singapore, Singapore 119077, Singapore. [5] These authors contributed equally: Zhihao Ren, Zixuan Zhang. ✉email: elelc@nus.edu.sg

Molecular identification of gases[1–5], liquids[6–8], and biomolecules[9–11] is a fundamental requirement for various applications such as environmental monitoring, healthcare, clinical diagnosis, and biological screening. Mid-infrared (MIR) fingerprint absorption, reflecting the generic information of molecule structures in chemical bonds and functional groups, provides natural optical probes for molecular identification. Harnessing vibrational fingerprint, IR spectroscopy offers a solution for non-invasive, non-destructive, label-free, and real-time recognition and monitoring of molecules, especially in the mixture. Furthermore, with the development of nanofabrication technology, artificially structured nanoantenna is demonstrated to enhance the IR fingerprint absorption by tailored plasmonic resonance[12,13]. This amplification effect caused by the strong plasmon-molecule coupling between plasmonic resonance and molecular vibration was well-explained by temporal coupled-mode theory (TCMT)[14,15].

To characterize the PNAs, sensitivity and bandwidth become two critical figure-of-merits (FOMs) that reflect the performance of PNA sensors[16]. However, the sensitivity and bandwidth of PNAs are limited by the small overlap between molecules and sensing hotspots and the sharp plasmonic resonance peaks. Since the plasmon-phonon coupling is induced by a localized electric field near the PNA surface, the straightforward approach is to concentrate the molecule to the active area, which is the hot spot of the electromagnetic field. The molecular enrichment strategies, including functionalized chemical bonding[17,18], chemical reaction[19,20], physical adsorption[21–23], optical trapping[24] as well as passive trapping by undercut structure[25–28] bring in the additional working requirements (e.g., chemical stimuli, temperature, pressure, pump power, etc.) for specific molecules, impeding the system for global molecular recognition. The other method to improve the sensitivity is to increase the intensity of the electric field by squeezing the gap between adjacent PNA into the nanometer scale[29–31]. However, the narrow gap ruins the sensing performance by a decrement in the active area and increases fabrication cost. Therefore, other approaches like hybrid 2D materials[32–34], device undercut[26,28], and homo/heterogeneous bonding[35–38] are developed to bypass the fabrication limitation and to achieve a large area of electromagnetic hot spots for sensing.

In addition to sensitivity, the detection range is another critical FOM of PNA sensors, reflecting the number of fingerprint absorption peaks that can be captured. Thanks to the sharp resonance peaks of PNA, the enhancement becomes the maximum only when molecule fingerprint absorption peaks match with the PNA resonance, which is a very narrow bandwidth. Therefore, to detect more absorption peaks in the MIR region, multi-resonant PNA sensors are proposed to collect broadband spectrum data to recognize lipids and proteins from separate absorption wavelengths[39]. Nevertheless, the individual resonances of PNA by a different order of resonance modes also fail to cover the whole spectrum of IR fingerprint wavelength region from 5.5 μm to 10 μm because of the gaps between two resonance peaks[40,41]. Therefore, to collect continuous spectral fingerprint absorption, pixelated all-dielectric nanoantenna array and tunable antenna by incident angle were proposed for ultra-broadband spectroscopic analysis for molecular barcode imaging and fingerprint absorption retrieving[42,43].

Thanks to the excellent field confinement at resonance wavelength, plasmonic nanoantennas (PNAs) also serve as an ultrasensitive refractometry sensor to capture the refractive index of analytes by wavelength shifts (e.g., color change in visible light), which carries the information of physical properties of molecules[44,45]. Unfortunately, in the IR fingerprint region, the PNA signal of absorption changes and wavelength shifts always

comes together, hindering the usage of complementary information about the analyte's physical and chemical properties for molecule identification. Artificial intelligence (AI) is a powerful tool for extracting the feature of data from different domains to achieve enhanced pattern recognition from IR spectra[46,47]. With the aid of machine learning (ML), decoupling the IR fingerprint absorption and refractive index change induced by molecules has been reported to monitor protein dynamics of secondary structure α-helix and β-sheet[48] and recognition of physiological biomarker of glucose and fructose[49]. However, the molecular recognition capabilities are limited to two specific molecules due to the narrow bandwidth at an operating wavelength of around 6 μm. Leveraging the multi-resonant PNAs, the deep learning method is proposed to augment the dynamics monitoring between four major classes of biomolecule (lipids, proteins, nucleic acids, and carbohydrates) from the absorption spectra in three working wavelengths of 3.4 μm, 6.5 μm, and 9 μm[50]. However, since the same classes of molecules always have a strong overlap in fingerprint absorption due to the similar chemical structure, it is challenging to distinguish the same classes of molecules in small concentrations by monitoring single absorption peak of each molecules. Therefore, new nanoantenna structures need to be developed for continuous broadband detection to capture multiple absorption peaks in fingerprint regions and new methods need to be conducted to distinguish the chemically similar molecules from the partially overlapped absorption spectra.

We propose a molecular identification platform by wavelength-multiplexed hook nanoantenna array (WMHNA) to enhance the sensitivity and detection bandwidth of PNA-based spectroscopy. In addition to field enhancement and molecule enrichment, we propose a loss engineering method to design HNAs by investigating another key parameter of damping rate, which influences the sensitivity of plasmon-molecules coupling. By tailoring the radiative to absorptive loss ratio, the sensitivity of molecular vibration detected by HNAs can be improved dramatically under the optimal condition supported by temporal coupled-mode theory (TCMT). Leveraging the gradient change of HNA dimensions, the WMHNA performs ultra-broadband working wavelengths ranging from 6 μm to 9 μm, matching with the MIR fingerprint region. Based on WMHNA, we propose a machine learning method to recognize molecules from WMHNA signals by decoupling the complementary physical (refractive index) and chemical (fingerprint absorption of the chemical bond) information. With the aid of principal component analysis (PCA) together with supporting vector machine (SVM), the proof-of-concept demonstration of molecular recognition from mixed alcohols (methanol, ethanol, and isopropanol) shows 100% identification accuracy. With the excellent sensing performance governed by loss engineering and wavelength multiplexing, the WMHNA platform paves the way to ultrasensitive on-chip molecular identification from various species and low concentrations for state-of-the-art chemical and biomolecular analysis.

## Results

**Working mechanisms of WMHNA microfluidic sensing platform**. The concept of the WMHNA microfluidic sensing platform is shown in Fig. 1. As shown in Fig. 1a, the WMHNA chip made by gold nanoantenna on an IR glass substrate-calcium difluoride (CaF$_2$) is bonded to a microfluidic chamber made by Poly-dimethylsiloxane (PDMS). The nanoantennas, which are physically in contact with fluidic analytes, perform as vibrational probes by light-matter interaction for molecular sensing. To excite the plasmonic resonance, the MIR light from an IR

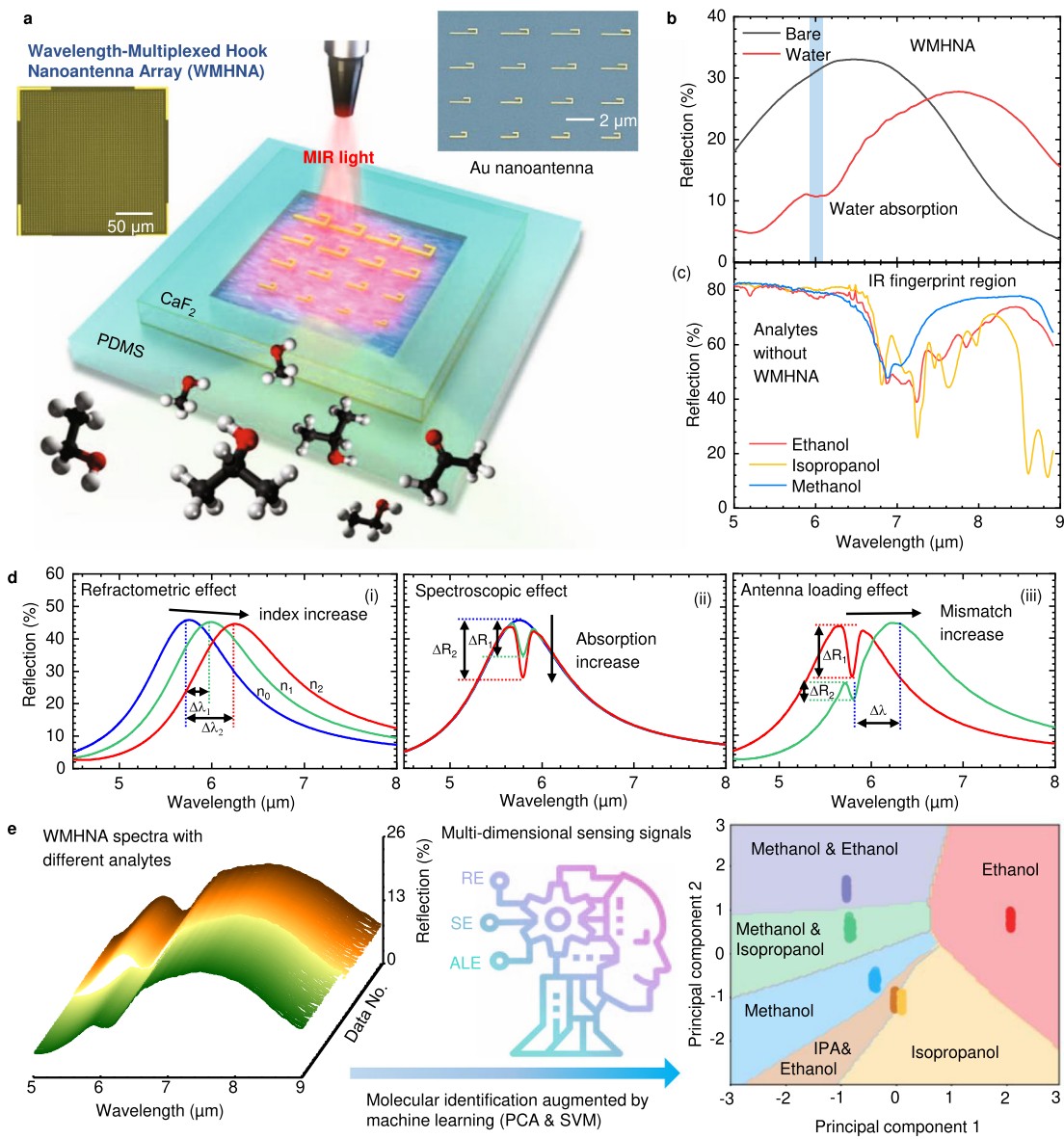

**Fig. 1 Working mechanisms of liquid molecular identification by wavelength-multiplexed hook nanoantenna array (WMHNA). a** Schematic drawing of WMHNA on CaF$_2$ platform with PDMS microfluidic chamber. Inset SEM image of one unit cell of WMHNA. **b** The far-field spectra of WMHNA with and without water measured in the reflection mode of the FTIR microscope. **c** The reference IR absorption spectra of alcoholic liquid (methanol, ethanol, and isopropanol) in the IR fingerprint region match with the broadband response of WMHNA. **d** Illustration of multi-dimensional information in WMHNA system. (i) Refractometric effect (RE): wavelength shift caused by the refractive index of analytes; (ii) Spectroscopic effect (SE): intensity drop due to the absorption of analytes; (iii) Antenna loading effect (ALE): the peak difference brought by the wavelength mismatch between analytes vibration and antenna resonance. **e** Machine learning process to extract multi-dimensional sensing information for recognition of 1% alcohols and their mixtures in water.

microscope with desired polarization states and incident angle shines on the backside of CaF$_2$ glass and transmits to nanoantennas. The reflected light is routed to an IR photodetector to capture the far-field spectral response from gold nanoantenna. The resonance wavelength of gold nanoantenna can be engineered by controlling the antenna length at the optical axis. When the resonance wavelength of the antenna is matched with molecular vibration, the enhanced absorption spectrum can be observed in a far-field response. To achieve a broadband enhancement, the WMHNA is designed by combining 16 nanoantenna structures with a gradient resonance wavelength thanks to the graded length. The far-field reflection spectrum is shown in Fig. 1b, covering a broad bandwidth from 6 to 9 μm in the IR fingerprint region. The broadband resonance peak of WMHNA is well-matched with the fingerprint absorption of

alcoholic liquids of methanol, ethanol, and IPA as shown in Fig. 1c. The fingerprint absorption spectra of alcoholic liquids are tested in a customized liquid cell with a gold mirror. In addition to the bandwidth, the hook nanoantenna also performs an enhanced sensitivity thanks to reducing radiative loss by folded structures. The detailed analysis and characterization will be conducted in the next section.

The light-matter interaction between nanoantennas and molecules can be distinguished with three significant effects – the refractometric effect (RE), spectroscopic effect (SE), and antenna loading effect (ALE). These effects happen simultaneously when molecules come into the ultra-confined electromagnetic field near the antenna surface in subwavelength scales. To demonstrate the individual response caused by three effects, we use finite-difference time-domain (FDTD) methods to

simulate the antenna response and manipulate the optical properties (refractive index, $n$ and absorption coefficient, $k$) of analytes by Lorentz model, which has been used in our previous works[19,51] (See supplementary Note S1 for details). Figure 1d shows the influence of antenna spectrum by RE, SE, and ALE individually. As shown in Fig. 1d-i, the refractometric effect, caused by the refractive index of analytes, brings a redshift of resonance wavelength as $n$ increases. Larger $n$ of analytes brings a longer resonance wavelength of nanoantennas due to the increment of effective optical length, proportional to the refractive index of surrounding environments and the physical length of nanoantennas. Figure 1d-ii shows the spectroscopic effect thanks to the plasmon-phonon coupling between antenna resonances and molecular vibrations. The resonance spectrum of nanoantenna is tailored by molecular absorption to form a dip into a reflection curve. The stronger molecular absorption, the more significant intensity drops in the reflection spectrum. According to Beer-Lambert's law, absorption strength is characterized by absorptance, which is proportional to the concentration of molecules and optical path length. For our microfluidic sensing system, the optical path length is fixed and determined by the nearfield confinement of nanoantenna, which is at an order of hundreds to thousands of nanometer scales. Therefore, the concentration of analytes can be read out from the intensity drops from the reflection spectrum. The antenna loading effect is caused by the wavelength mismatch between antenna resonance and molecular vibration, as shown in Fig. 1d-iii. For two molecules with the same absorption strength, small intensity drops can be read out when the wavelength mismatch increase, indicating a larger refractive index. The detailed analysis and characterizations of ALE are included in the following section. In real sensing applications, the $n$ and $k$ of analytes always vary simultaneously when changing the concentration of any molecules, resulting in a complex change of spectrum with the combination of RE, SE, and ALE. Therefore, we define these complex changes as multi-dimensional sensing signals.

We propose a machine learning algorithm using principal component analysis and support vector machine to extract the individual features from the multi-dimensional sensing signals. As shown in Fig. 1e, a group of spectra data from WMHNA with different analytes is selected as inputs of the machine learning algorithm. With the feature extraction by PCA, the sensing signal with different degree-of-freedom are identified as different principal components (PCs), indicating the influence from RE, SE, and ALE. The relationship between PCs and sensing features from each effect is further discussed in the later section. Then the SVM algorithm is used to classify the data in the PC domain. The identification results of three major types of alcohols mixed in water with 1% concentration are shown in Fig. 1e. Each point in the PC domain refers to one spectrum data from WMHNA, and the color area indicates an obvious decision boundary of each type of target.

**Design principles of hook nanoantenna.** The resonant nanoantenna interacts with the molecular vibration at a matched wavelength with the plasmon-phonon coupling, enhancing the fingerprint absorption. In the coupling system, the radiative and absorptive loss play essential roles for the far-field spectrum, which is the sensing signal in our platform. Hook nanoantenna (HNA) provides an approach to manipulate the radiative loss by folding the gold nanorod structures, bringing in another degree of freedom to optimize the sensor performance.

To understand the design principles, we use the TCMT model[52] to analyze the plasmon-molecules coupling system. Based on TCMT, we can get equations for coupling system[51] and

derive the transmission and reflection spectral dispersion as (detailed equation derivation is shown in Note S1)

$$T(\omega) = \left|\frac{S_t}{S_{in}}\right|^2 = \left|\frac{j(\omega - \omega_0) + \gamma_a + \frac{\mu^2}{j(\omega - \omega_m) + \gamma_m}}{j(\omega - \omega_0) + (\gamma_a + \gamma_r) + \frac{\mu^2}{j(\omega - \omega_m) + \gamma_m}}\right|^2 \quad (1)$$

$$R(\omega) = \left|\frac{S_r}{S_{in}}\right|^2 = \left|\frac{\gamma_r}{j(\omega - \omega_0) + (\gamma_a + \gamma_r) + \frac{\mu^2}{j(\omega - \omega_m) + \gamma_m}}\right|^2 \quad (2)$$

where $\omega_0$ and $\omega_m$ represent the angular frequency of resonance for HNA and molecular vibration, respectively. $\gamma_a$ and $\gamma_r$ denote the radiative and absorptive losses of HNA, while $\gamma_m$ is the absorptive loss of molecules. $\mu$ is the coupling strength between HNA and molecular vibration. From Eqs. 1, 2, the enhanced vibration signal can be observed as a Fano-like line shape, a noticeable dip in HNA resonance when two resonance modes are well-matched ($\omega_0 = \omega_m$). The enhancement of the sensing signal is observed in the change of transmission or reflection intensity compared with intrinsic molecule absorption. Furthermore, the sensitivity of plasmonic sensors is defined as the intensity change of resonance spectrum of transmission ($\Delta T$) or reflection ($\Delta R$) and is expressed as (detailed equation derivation is shown in Note S1)

$$\triangle T(\omega = \omega_0) = T(\omega = \omega_0) - T|_{\mu=0}(\omega = \omega_0) = \frac{2\mu^2}{\gamma_a \gamma_m}\frac{f}{(1+f)^3} \quad (3)$$

$$\triangle R(\omega = \omega_0) = R(\omega = \omega_0) - R|_{\mu=0}(\omega = \omega_0) = -\frac{2\mu^2}{\gamma_a \gamma_m}\frac{f^2}{(1+f)^3} \quad (4)$$

where $\mu$ and $f$ denote coupling efficiency between HNA and molecular vibration as well as the ratio ($\gamma_r / \gamma_a$) between radiative ($\gamma_a$) and absorptive ($\gamma_r$) damping rate of the HNA, respectively. From Eqs. 3 and 4, we can find that $\mu$ and $f$ are two degrees of freedom to optimize the sensing performance, highlighted in Fig. 2a. The coupling efficiency is related to the nearfield intensity and the overlapping of molecules to the antenna nearfield. Figure 2b summarizes three approaches to optimize nanoantenna sensing by increasing $\mu$. Hotspots release is proposed to increase accessible sensing areas for analytes, which are usually blocked by the substrate, to further enhance the coupling efficiency[25–28]. Thanks to the ultra-confined electric field of PNAs, the coupling region only covers hundreds of nanometers near the antenna surface. Therefore, the molecule enrichment method is utilized to accumulate localized molecules in the effective sensing area[17–24]. Another approach to enhance the nearfield coupling is to increase the electric field intensity by squeezing the adjacent nanoantenna into the nanogap[29–31]. In this case, the narrower the nanogap is, the stronger the PNAs enhancement behave. However, all of these three methods introduce extra processes or materials to make the plasmonic sensors, which increase the fabrication cost or limit the working condition.

In addition to the near-filed coupling, engineering the antenna loss is another degree of freedom to manipulate the sensing signals, as illustrated in Fig. 2a. Figure 2c shows the influence of antenna loss on the sensitivity in transmission and reflection modes. The maximum sensitivity occurs when $f$ reaches 0.5 and 2 for transmission and reflection modes, respectively. The $\gamma_a$ is related to the ohmic loss of plasmonic material (e.g., Au in this work) and is almost the same among different antenna structures. Therefore, the philosophy to use hook shape in nanoantenna design is to engineer $\gamma_r$ to tune the radiation from electron oscillation by inducing inverse current from dipole resonance. The method to control radiation capability from the ratio of inverse current is merely adjusting the geometric difference ($\Delta L$)

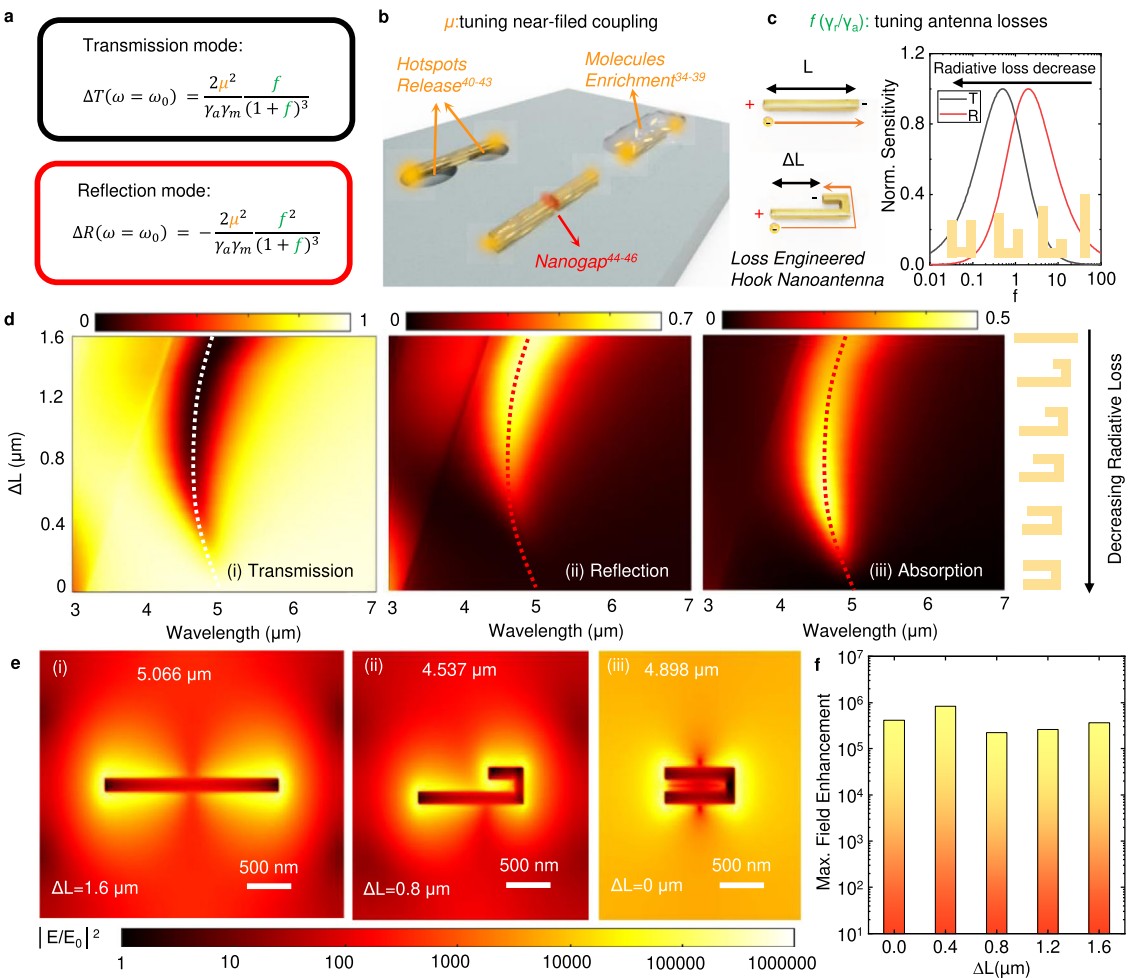

**Fig. 2 Design principle of hook nanoantenna. a** The derived intensity changes in transmission and reflection mode with the perfect match of molecular absorption and antenna resonance by TCMT. **b** The summary of methods affecting the coupling strength between molecules and antenna by previous works. **c** The proposed hook antenna for sensing optimization by tuning the radiative loss and the effect of loss ratio to intensity change in transmission and reflection mode. **d** The simulated far-field spectra in transmission (i), reflection (ii), and absorption(iii) of hook antenna by changing the ΔL, indicating the tuning of radiative loss. **e** The electric field enhancement distribution of hook nanoantenna devices at resonance wavelength with different ΔL of 1.6 μm (i), 0.8 μm (ii) and 0 μm (iii). (**f**) The extracted maximum electric field enhancement of hook nanoantenna devices with different ΔL.

between the long arm and short arm of HNA, as illustrated in Fig. 2c. With the design of hook nanoantenna, the radiative loss can be continuously tuned by controlling the folding degree of HNA, which is denoted as ΔL. By decreasing the ΔL, the radiative loss decreases due to the increased length of overlapping inverse current in the short arm of HNA, as shown in Fig. 2c. With the precise design of HNA, the optimal sensitivity of transmission and reflection modes can be achieved by different ΔL.

The FDTD simulation results of HNA with the change of ΔL are shown in Fig. 2d-f showing the optical properties of dipolar resonance. The connection of two arms of HNA only affects the resonance wavelength and is defined as a fixed value (400 nm) to fit our fabrication resolution. With the decrease of ΔL, both T (Fig. 2d-i) and R (Fig. 2d-ii) intensities drop, which means the antenna becomes less radiative (darker). However, the absorption signal reaches a peak value when ΔL equals 0.6 μm, which means the critical coupled point ($\gamma_a = \gamma_r$) of the HNA resonator. The nearfield distributions at resonance wavelength are shown in Fig. 2e. The electric field enhancement with no fold (ΔL = L = 1.6 μm), half fold (ΔL = 0.5 L = 1.6 μm), and full fold (ΔL = 0) are plotted in Fig. 2e-i-iii. The detailed nearfield profile of electric field polarity and magnetic field distraction are shown in Supplementary Note S2. To compare the field enhancement of

each hook antenna device, the maximum field enhancement is extracted in Fig. 2f. The results show that the enhancement of the electric field intensity of all HNA devices is at the order of $10^5$, and the maximum enhancement of the electric field intensity reaches 832891 times. There is a slight increment of the intensity (~2 times) from half to full fold of HNAs due to the superposition of the electric field near the two poles of the HNAs. However, compared to the dramatic change of radiative loss, the loss ratio (f) is a more dominant affecting parameter in different HNAs.

**Sensing characterization of hook nanoantennas.** We fabricate the gold HNA devices on $CaF_2$ substrate by a lift-off process with the poly(methyl methacrylate) (PMMA) resist patterned by electron-beam lithography. The detailed fabrication process is described in Methods. The optical microscope (OM), SEM, and atomic force microscope (AFM) images of fabricated HNA devices are shown in Supplementary Note S3. The Fourier-transform infrared (FTIR) spectroscopic microscope is used to measure the far-field response of the nanoantenna array. The detailed experimental setup is described in Methods. To characterize the sensing performance of each hook antenna device, 10 nm PMMA thin film is coated on top of HNA sensors.

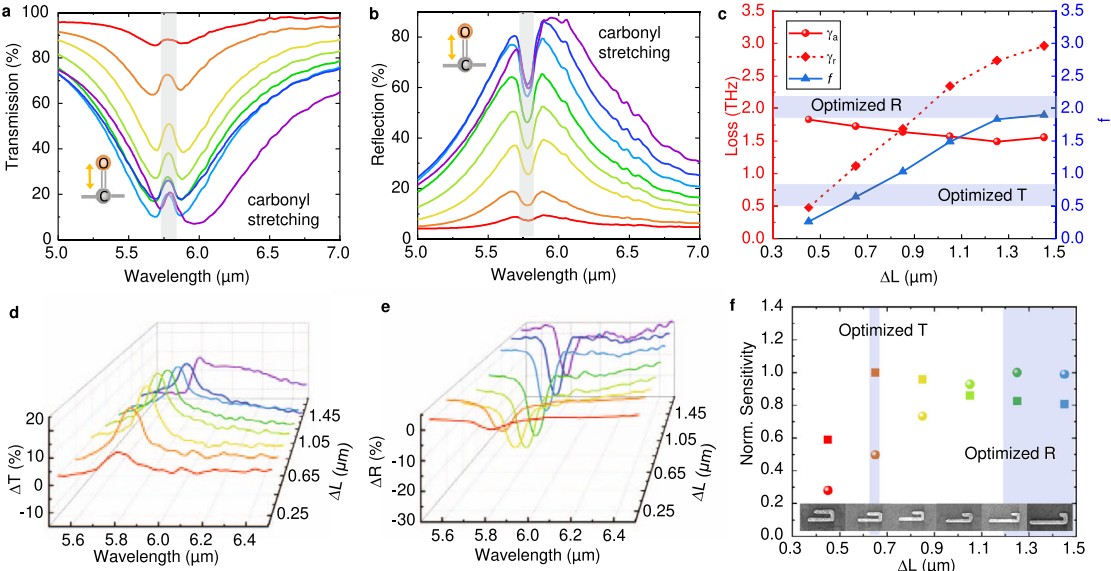

**Fig. 3 Experiment characterization of hook nanoantennas sensing performance. a, b** The testing transmission (**a**) and reflection (**b**) spectra of hook nanoantenna devices with different ΔL for PMMA thin film sensing. The carbonyl stretching vibration in 5.76 μm or 1736 cm$^{-1}$ is used to characterize the sensitivity of hook nanoantennas. The colors of the line refer to the different ΔL in **e, f. c** The extracted absorptive and radiative loss of hook nanoantennas by fitting the experimental spectra with TCMT. **d, e** The baselined corrected spectra difference in transmission(**d**) and reflection(**e**). **f** The calculated sensitivity of hook nanoantenna devices with different ΔL. The sphere symbols refer to reflection mode and square symbols represent transmission mode.

We fabricated a group of 6 HNA devices with different ΔL to show the effect of radiative loss on the sensitivity of HNA sensors. A substantial intensity change caused by the carbonyl stretching is observed at ~5.8 μm in transmission (Fig. 3a) and reflection (Fig. 3b) mode. To obtain the radiative and absorptive loss of HNA devices, we fit the transmission and reflection spectra without PMMA cladding using Equation S9, 10. As shown in Fig. 3c, the radiative loss decreases as ΔL decreases, but the absorptive loss is almost unchanged, resulting in a small $f$ with short ΔL. The extracted difference signals are plotted in Fig. 3d, e by normalizing to the baseline curves, which are obtained from the asymmetric least-squares smoothing (AsLSS) algorithm[53] to the sensing spectra of HNA with different ΔL. The sensing signal can then be characterized by the intensity change of transmission and reflection spectra extracted in Fig. 3f. From this figure, we observe that the highest sensitivity of transmission mode (ΔT) comes when ΔL equals 0.65 μm, while 1.25 μm for reflection mode (ΔR). As ΔL decreases, $\mu$ remains almost unchanged because of the similar intensity of near field (Fig. 2f), but $f$ decreases due to the reduced $\gamma_r$ caused by the short electrical length of the antenna and the similar $\gamma_a$ caused by the same antenna length. The extracted loss ratio of the two optimized devices is 0.644 (ΔL = 0.65 μm) and 1.834 (ΔL = 1.25 μm), which agrees with the theoretical prediction of optimal condition, which is $f$ equals 0.5 for transmission mode, and $f$ equals 2 for reflection mode.

**Wavelength multiplexed hook nanoantenna array**. To illustrate the purpose of wavelength-multiplexed designs, we need to figure out the antenna loading effect first. As mentioned in Eq. 1, the resonance wavelength of nanoantenna and molecules play significant roles in the spectral line shape. When the two resonance wavelengths are well-matched, the resonance spectrum performs a Fano-like dip. In contrast, an asymmetric change of spectrum happens when the two resonance wavelengths are mismatched. We fabricate a group of HNA devices with different resonance wavelengths by changing the L of HNA to characterize the ALE

on 10 nm PMMA thin film. In Fig. 4a, b, the highest sensitivity is achieved when the resonance wavelengths of HNA and molecules are well-matched (Fig. 4c). A two times improvement of sensitivity is observed by the HNA with 2.22 μm length compared with the HNA with 2.4 μm length. After optimization, the arm length ratio is fixed at 1:3 to achieve the highest sensitivity at reflection mode. Therefore, to simultaneously achieve the best sensitivity and broad bandwidth, the wavelength-multiplexed structures are designed to have the wavelength-scalable response by gradually increasing the total length with the fixed folding degree of HNA.

The spectrum of the 16-element WMHNA is shown in Fig. 4d, showing the wavelength-multiplexed response from ~5 μm to ~7.8 μm by changing the L. To compare the sensing performance of WMHNA, we spin coat two types of molecules (silk protein and PMMA) separately on WMHNA. Figure 4b shows the sensing results of silk and PMMA on HNA supercell with the broadband response from ~5.5 μm to ~8.5 μm. Multiple fingerprint absorption peaks, labeled in Fig. 4f, are captured by the broadband device. The redshift of the HNA supercell spectrum is caused by the effect induced by the refractive index of analytes indicating the RE of WMHNA. We further compared the sensing performance with selected HNA elements (P1, P8, and P16) from WMHNA, showing that WMHNA better enhances multiple absorption peaks from broad wavelength ranges, while HNA only reaches the best enhancement at narrow wavelength ranges near resonance wavelengths. Both WMHNA and HNA perform the significant enhancement (3 orders of magnitude) of absorption spectrum with the direct measurement of thin-film without nanoantenna. However, WMHNA performs less influence from ALE in the broadband wavelength range, showing good performance for sensing broadband fingerprint absorption.

**Liquid dynamics monitoring with broadband fingerprint absorption**. To demonstrate the capability of liquid dynamics monitoring with broadband fingerprint absorption, we integrate WMHNA into a microfluidic system (Fig. 5a), which is compatible with biomolecular systems for molecular detection in the

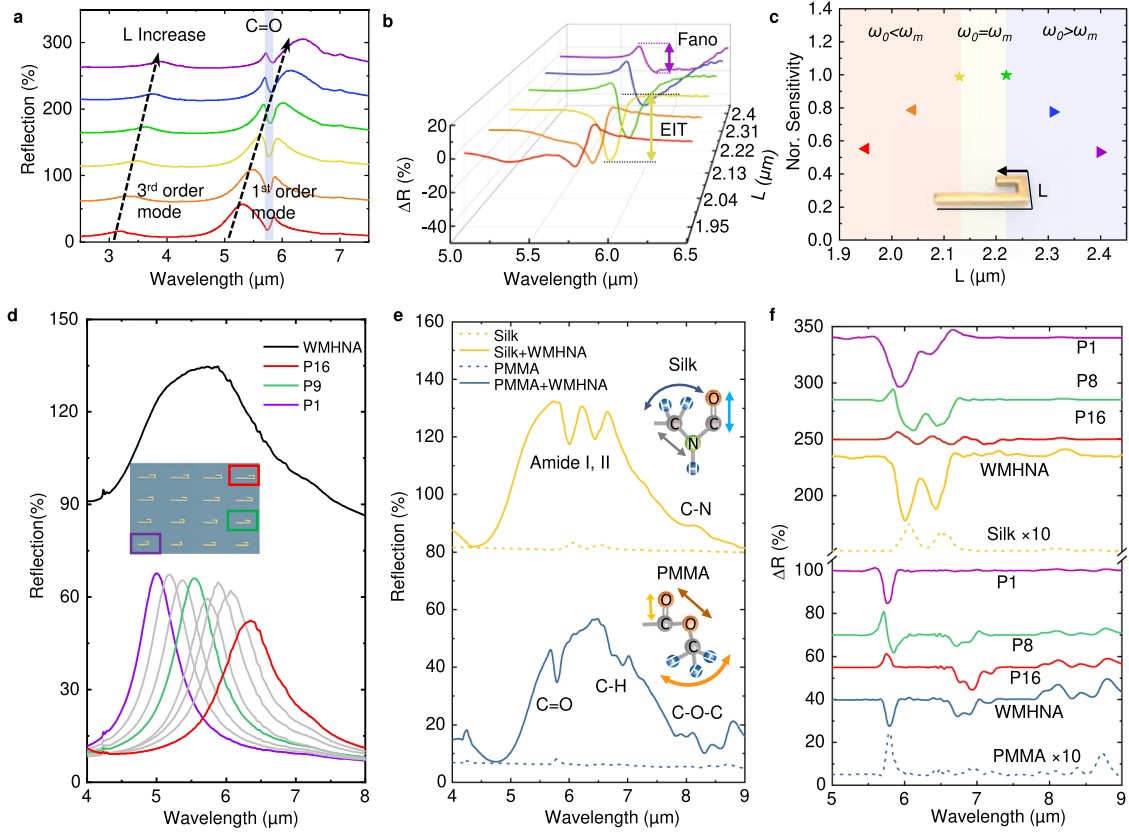

**Fig. 4 Experiment characterization of WMHNA to decrease loading effect with broadband absorption enhancement. a** The testing reflection spectra of hook nanoantenna with PMMA thin film at different L. **b** The baseline-corrected reflection spectra from (**a**). **c** The normalized sensitivity of hook nanoantenna devices with different L indicates the loading effect. Sensitivity becomes maximum when the two wavelengths of molecular vibration and antenna resonance are matched. **d** the reflection spectra of WHHNA and hook nanoantenna devices (P1, P8, and P16) to form the supercell. **e** The sensing characterization of WMHNA by two types of thin films of PMMA and silk. **f** Baseline-corrected sensing signal of WMHNA compared with device P1, P8, and P16 by two analytes of PMMA and silk.

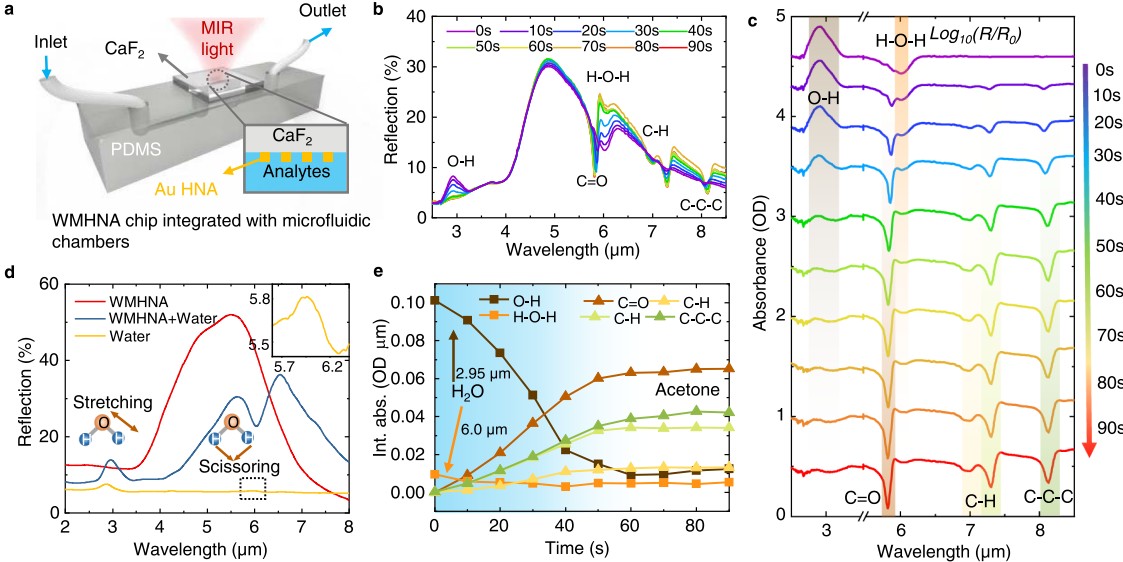

**Fig. 5 Demonstration of broadband liquid dynamics monitoring of acetone and water using WMHNA integrated with microfluidics. a** The schematic drawing of WMHNA chip integrated with PDMS microfluidic chamber. **b** The dynamic monitoring of broadband spectra of WMHNA with acetone injection to water. **c** The extracted absorbance spectra at different time. **d** The comparison of spectra of WMHNA with and without water. **e** The integrated absorbance at each absorption peak indicates the dynamic change of analytes.

aqueous environment. The microfluidic channel is formed by a 3D-printed mold and is fixed on a microscope slide. The IR light transmits through the CaF$_2$ substrate to the nanoantenna and reflects the photodetector. The response of WMHNA for water with the enhancement of stretching absorption at 2.95 μm and scissoring absorption at 6.0 μm is shown in Fig. 5d. The stretching mode is in the non-resonant regime of WMHNA, showing an enhancement factor of 4.17 thanks to the localized electric field. In the resonant regime, an improvement of 34.36 times of reflection change is observed for the scissoring mode.

Furthermore, the absorption of water in IR has a minimal influence on the nanoantenna spectrum apart from the molecular vibration regime, allowing the liquid recognition with water-based solvent, which is so absorptive in traditional IR spectroscopy that the intensity of light becomes very weak to identify molecules. The secret is that the ultra-confined electromagnetic field near the antenna surface shortens the effective optical path length to subwavelength scales. Therefore, a slight change of intensity of light is observed according to Beer-Lambert's law.

We also perform the dynamic monitoring of acetone in water to mimic the real-time dynamic monitoring of metabolic in biological samples. As shown in Fig. 5b, the real-time spectrum indicates the analyte change at the WMHNA surface as time goes by. Each curve indicates the real-time spectrum of a mixed solvent of acetone and water, reflecting in-situ concentration information of acetone and water and dynamic change versus time. Multiple fingerprint absorption peaks are captured to have rich information of chemical bond changes. (Fig. 5b) To further characterize the dynamic change using fingerprint absorption, we calculate the absorbance of each spectrum using logarithms from the reflection difference from the baseline signal. The baseline-corrected absorbance spectrum at a different time is shown in Fig. 5c in broad wavelengths range from 2.5 to 3.5 μm for O-H bond of water at 2.95 μm and 5.5 to 9 μm for various fingerprint peaks including H-O-H bond for H$_2$O at 6.0 μm, C=O, C-H, C-C-C bond for acetone at 5.7 μm, 7.0 and 7.3 μm, 8.3 μm, respectively. By integrating the absorbance spectrum, the dynamic behavior of water and acetone can be monitored by changing different chemical bonds (Fig. 5e). The reduction of O-H and H-O-H bond absorption at 2.95 μm and 6.0 μm represents the decrease of water concentration, while the increasing of C=O, C-H, C-C-C bond absorptance at 5.7 μm, 7.0 and 7.3 μm, 8.3 μm indicate the introduction of acetone molecules into the microfluidic system. Thanks to the enhancement of vibration absorption and the reduction of optical path length, our WMHNA platform is suitable for liquid sample analysis using water as a common solvent, paving the way to protentional application in biological sample screening.

**Molecular identification by machine learning**. To demonstrate the molecular identification properties of WMHNA, we select three types of chemically similar alcoholic liquid-methanol, ethanol, and isopropanol. Both of the molecules have the same functional group of hydroxy and methyl bond, resulting in similar absorption spectra in 6 μm to 9 μm wavelengths. Therefore, it is not easy to distinguish them in a mixture with a narrowband HNA. We designed a series of experiments to characterize the recognition capability of WMHNA using 1% methanol, ethanol, and IPA in water and mixture sets of each two in the same volume. With the injection of liquid from microfluidics, the response of WMHNA is plotted in Fig. 6a. The apparent dips of the reflection spectrum at 6.0 μm are induced by water scissoring absorption. The fingerprint absorption of small molecule alcohols is captured from 6.5 μm to 9 μm and is extracted from the

WMHNA spectrum in Fig. 6b. Due to the low concentration of analytes, the change of reflection at absorption is small and cannot be detected without HNA. To process the small signal, we applied a second derivative to extract the characteristic of each spectrum from the HNA supercell (Fig. 6c), which is widely used in traditional IR spectroscopy analysis. However, it is still difficult to distinguish clearly with the classic data processing methods from the enhanced spectrum of HNA by solely analyzing the fingerprint absorption. Therefore, we propose an ML method using PCA to process the HNA data for extraction of multi-dimensional information from WMHNA, which is absorption peaks induced by vibration of the chemical bond, the wavelength shift of HNA resonance induced by the refractive index of molecules, and the intensity change of water absorption induced by ALE of wavelength detuning.

The results of PCA processed spectra are shown in Fig. 6e by dimension reduction to three principal components (PC) axes. The multi-dimensional sensing signal is extracted from the spectrum in the PC domain. The first PC represents the wavelength shift of HNA resonance by RE and the second PC represents the modulation of water absorption peak by ALE. The third PC represents the fingerprint absorption of three molecules by SE. The order of PC represents the degree of difference between each spectrum. In the 3D PC space shown in Fig. 6f, each point represents the spectrum data from WMHNA, and each cluster represents one type of molecule combination.

Furthermore, we use machine learning algorithms for this molecular identification problem. Support vector machine (SVM) is a supervised machine learning model that uses classification algorithms, which is suitable for our application due to its conciseness and low computational cost. During the training of classifiers, an SVM model takes points in multi-dimensional space and outputs the hyperplanes that best separate the point clusters. The SVM classifies it when given test data by comparing its location in multi-dimensional space with the hyperplanes. Because SVM is a classification algorithm for two-group classification problems, we transform it into a set of binary classification problems to distinguish 6 different gases. The whole dataset from WMHNA is 50 spectra for each molecule, and we divide it as a training dataset (40) and a testing dataset (10). Firstly, we use the first two PCs to train the classifiers and evaluate the results. The SVM kernel for this model is Radial Basis Function (RBF) kernel. Finally, the accuracy of this classifier for both the training set and test set is 100%. It can be seen that the classifier trained with two PCs works well, with the different gases correctly identified and the clusters being well contained in the appropriate regions.

With the help of ML, the IR spectrum of HNA with different molecules can be reduced to three principal components, which indicate the three key features of loading effect, wavelength shifts, and enhanced fingerprint absorption. With the full utilization of multiple dimension information, the recognition becomes more efficient by monitoring the complementary physical (refractive index) and chemical properties (absorption fingerprints) of molecules, bringing in another degree of freedom into IR spectroscopy analysis by refractometry and plasmonic properties. Compared with previous literature that demonstrates the identification of two molecules mixture by monitoring two absorption peaks[49], our work demonstrated simultaneously monitoring of 15 absorption peaks and used to identify three molecules mixture. Furthermore, with the aid of dimension reduction by PCA, the multi-dimensional information from HNA is easily decoupled and analyzed, paving the way to achieve global molecular identification and real-time monitoring by training with deep neural networks (DNN).

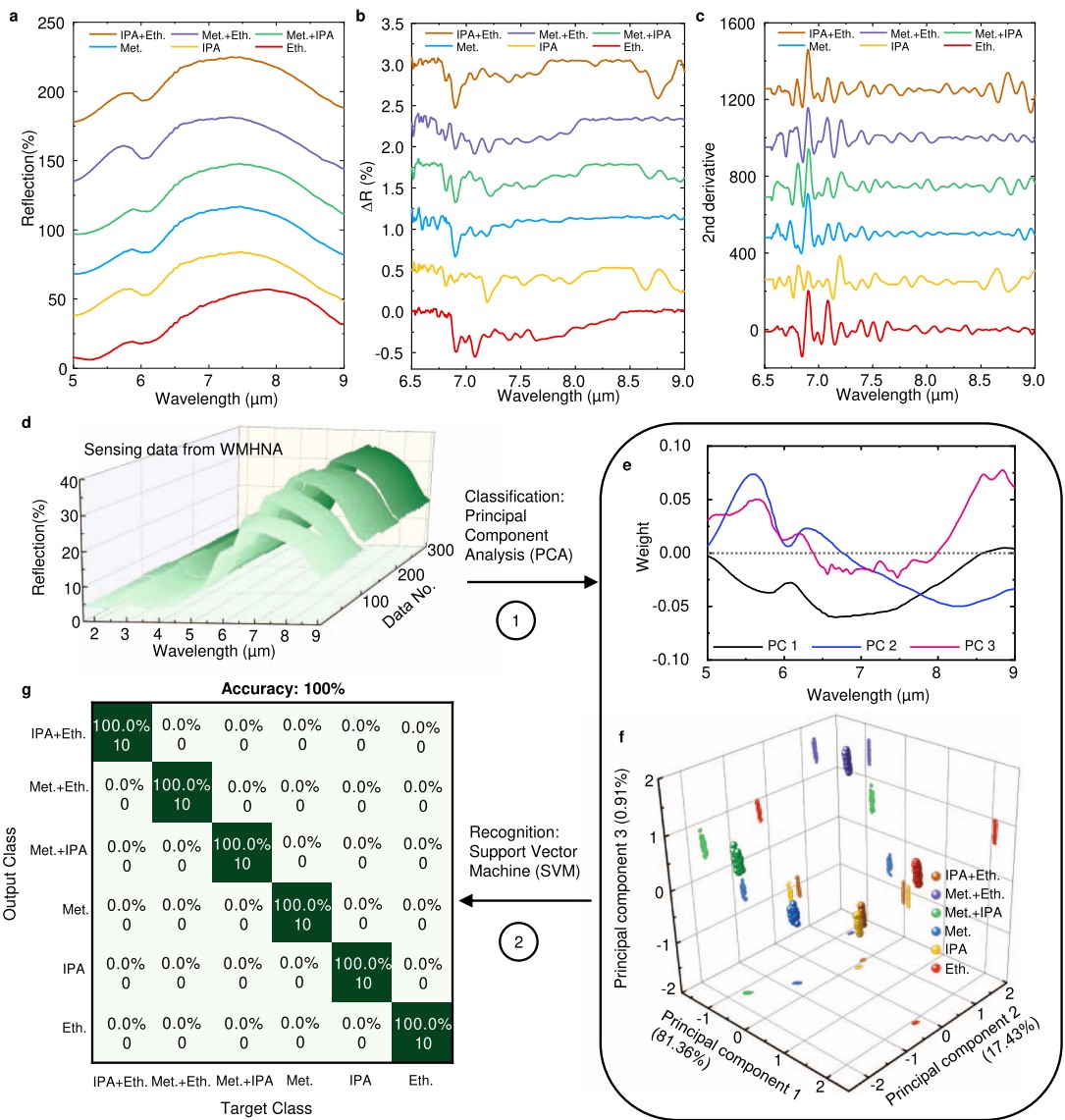

**Fig. 6 Demonstration of molecular identification of low concentration alcohols with multi-dimensional sensing signal from WMHNA by machine learning. a** The broadband spectra of WMHNA under different analytes. All alcohol solvents are diluted to 1% in DI water. **b** The extracted absorption spectra of different alcohol solvents from WMHNA. **c** The corresponding second-order derivative of absorption spectra in (**b**). **d** The reflection spectra of sensing data with different analytes states. **e** The machine learning processed spectra of WMHNA after dimension reduction by principal component analysis. the first principal component (PC 1) represents the antenna loading effect of water absorption peaks at 6.0 μm. The second principal component (PC 2) represents the wavelength shift of WMHNA due to the refractive index of the analyte. The third principal component (PC 3) represents the fingerprint absorption of molecules. **f** The weight of scores of each spectrum in three-dimensional space after PCA for WMHNA. Each cluster indicates one type of molecule and its mixtures. **g** The confusion map for machine learning outcome indicates the 100% accuracy of molecular identification.

## Discussion

In PNA-based surface-enhanced IR absorption spectroscopy (SEIRAS), the great challenge is the limited bandwidth of nanoantenna resonances compared with the ultra-broadband vibrational absorption fingerprints. Previous works show many solutions using multiple resonances of nanoantenna structures to capture the plasmonic enhanced molecular fingerprints in separate wavelength ranges for sensing different molecule species[22,39,41,50,54–56]. We propose hook nanoantenna supercell with wavelength multiplexing by the gradient change of antenna length of 16 elements to extract molecular absorption peaks in IR fingerprint regions from 6 to 9 μm wavelengths. The sensing performance of WMHNA is characterized by thin-film coating (PMMA and silk) and microfluidic dynamics (acetone and water)

as a proof-of-concept. On one hand, with the reverse current induced in HNA, the sensitivity of HNA is improved by engineering the radiative loss to achieve the optimized loss ratio. On the other hand, compared with the single nanoantenna designs with low quality factors, the wavelength of molecular absorption is matched with the specific hook antenna elements in WMHNA, maximizing the sensitivity caused by antenna loading effects.

Furthermore, based on the massive spectral data collected by the WMHNA-SEIRAS platform, we develop machine-learning-enabled spectroscopy to extract the multi-dimensional features from refractometric effect, spectroscopic effect, and antenna loading effect. Thanks to the high sensitivity and broad bandwidth, molecular identification can be achieved from the complementary physical (refractive index) and chemical (absorption

fingerprint) properties of molecules. With the mixture of diluted alcoholic solutions, the microfluidic-integrated WMHNA captures 15 absorption peaks of methanol, ethanol, isopropanol, and water in 6 to 9 μm wavelengths. Additionally, with the aid of a machine learning algorithm (PCA and SVM), the multi-dimensional information can be classified effectively, resulting in 100% accuracy from random 4:1 data spiting for training and recognition sets. Using the design principle of wavelength multiplexing, the WMHNA can be designed to cover all IR fingerprint ranges by engineering the antenna length for specific molecular monitoring like proteins[48], sugars[49], lipids[17], nucleic acids[57], and volatile organic compounds (VOCs)[58]. Our work brings deep insights into machine-learning-enabled IR spectroscopy technologies for small-volume, real-time, ultra-sensitive, in-vitro molecular dynamic analysis in the aqueous environment, paving the way to the state-of-art applications of drug screening[59], clinical diagnosis[60], healthcare[61], and environmental monitoring[62].

## Methods

**TCMT modeling**. The temporal coupled-mode theory (TCMT) is used to model the coupling behavior between PNA and molecular vibration. We treat the plasmonic resonant (denoted as P) as a bright mode that is coupled to the incident light, while we treat the molecular vibration (denoted as M) as a dark mode, in which coupling efficiency is much lower than PNA and can be ignored in their coupling system. The detailed derivation of equations is shown in supporting information. The MATLAB codes are generated to fit the simulation and experiment spectra with derived equations.

**FDTD simulation**. The finite-difference time-domain (FDTD) method (Lumerical FDTD) is performed to simulate the far-field spectrum and the nearfield distribution of plasmonic hook nanoantennas. The light source is selected as a plane wave to simulate the incidence of light from free space. The incidence angle and polarization state are adjusted to the desired orientation to excite the dipolar mode plasmonic resonance of HNA. The refractive index of $CaF_2$ is set at 1.38 at wavelengths ranging from 2 μm to 10 μm. The periodic boundary at the x and y-axis (Fig. 2a) is selected to simulate the effect of the periodic antenna array, and the PML boundary is chosen at the z-axis to transport light into free space. The minimum mesh size is set to be 20 nm at X and Y direction and 10 nm at Z direction, which is 10 times smaller than the smallest dimension of nanoantennas. There is also a trade-off between simulation accuracy and simulation time. When we further decrease the mesh size, the simulation time increases dramatically, while the results remain almost the same.

**Nanoantenna fabrication**. For the fabrication of HNA, electron-beam lithography (EBL, Jeol 6500FS) and the lift-off process are used to pattern the nanometer scale gold structure. Before EBL, the $CaF_2$ chip was firstly rinsed by Acetone and IPA solutions for 1 min with sonication. After that, the chip is treated under oxygen plasma for the uniform formation of PMMA 495 K A5 photoresist, which is spin-coated at 4000 rpm for 1 min. Since the conductivity of the $CaF_2$ chip is low, an additional E-spacer layer is spin-coated at 2000 rpm for 1 min to avoid charge accumulation during EBL. After EBL, the development with 30 s using PMMA developer (MIBK: IPA=1:3) is used to remove PMMA resist under exposure following by the cleaning with IPA for 30 s. Then electron beam evaporation (AJA International Inc.) is proceeded to deposit 80 nm thick gold on top of $CaF_2$ substrate and PMMA photoresist. To lift off the nanoantenna pattern, the chip is placed in acetone for one day and rinsed by IPA.

**FTIR measurement**. A Fourier-transformed IR (FTIR) microscope (Agilent Cary 660) with an FTIR spectrometer (Agilent Cary 620) and liquid-nitrogen-cooled HgCdTe (mercury cadmium telluride, MCT) detector is used to characterize the spectral response of hook nanoantenna. The background signal is collected from the $CaF_2$ chip using 16-32 scans at 8 $cm^{-1}$ resolution to compensate for the MIR gas absorption (mainly water vapor and $CO_2$) from the ambient. Then the sample scan is performed using 16-32 scans at 8 $cm^{-1}$ resolution to capture the spectral response of nanoantenna. The scanning area is adjusted to 200*200 $μm^2$ to fit the nanoantenna area. For liquid sensing, a microfluidic chamber made by PDMS is bonded to a $CaF_2$ chip to allow the contact of the liquid analyte with HNA, and the spectrum is captured simultaneously.

**Machine learning by PCA and SVM**. The principal component analysis is used in exploratory data analysis and for making predictive models. It is commonly used for dimensionality reduction by projecting each data point onto only the first few principal components to obtain lower-dimensional data while preserving as much of the data's variation as possible. To facilitate visualization of the feature space, PCA was performed in MATLAB_R2020a. In this case, a covariance matrix was computed using a factorization of singular value decomposition (SVD) for the normalized set of features from which the eigenvectors and eigenvalues were extracted. Each principal component was constructed as a linear combination of the initial features. The first three principal components were then used to display 3D scatter plots of the features.

A support vector machine constructs a hyper-plane or set of hyper-planes in a high or infinite-dimensional space, which can be used for classification, regression, or other tasks. Intuitively, a good separation is achieved by the hyper-plane that has the largest distance to the nearest training data points of any class, since in general the larger the margin the lower the generalization error of the classifier. The proposed SVM classifiers were developed on Python 3.6 using the sci-kit-learn package. When training an SVM with the Radial Basis Function (RBF) kernel, we need to consider the two parameters C and gamma. The gamma parameter defines how far the influence of a single training example reaches. The larger gamma is, the closer other examples must be affected. The C parameter trades off the correct classification of training examples against the maximization of the decision function's margin. A low C makes the decision surface smooth, while a high C aims at classifying all training examples correctly. gamma defines how much influence a single training example has. Here we set the C parameter as 1.0 and the gamma parameter as 0.1.

## Data availability

All technical details for producing the figures are enclosed in the supplementary information. Data are available from the corresponding author C.L. upon reasonable request.

## Code availability

The codes that support the findings of this study are available from the corresponding author C.L. upon reasonable request.

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

## Acknowledgements

This work was supported by Advanced Research and Technology Innovation Centre (ARTIC) under the research grant of R-261-518-009-720/A-0005947-20-00; National Research Foundation (NRF) under the CRP-15th research grant of NRF-CRP15-2015-02 project; Agency for Science, Technology and Research (A*STAR) under the RIE 2020 research grant of A18A5b0056 project; Ministry of Education (MOE) under the research grant of R-263-000-E14-114/ A-0005138-01-00 at National University of Singapore.

## Author contributions

Z.R. and C.L. generated the design concept of WMHNA. Z.R. performed antenna design, fabrication, and FTIR testing. Z.Z. performed machine learning algorithm. J.W. conducted the theoretical model and advised on fabrication and FTIR testing. Z.R., Z.Z., and B.D. analyze the WMHNA data. C. L. supervised the project. All authors prepared the manuscript and made revision to the peer reviews.

## Competing interests

The authors declare the following competing interests: Z.R. and C.L. are inventors on the international patent (WO 2022/045970 A1) submitted by National University of Singapore that covers design methodologies and machine learning applications of HNA and WMHNA. The remaining authors declare no competing interests.
