## [Peer Review File · Nature Communications]

Wavelength-Multiplexed Hook Nanoantennas for Machine Learning Enabled Mid-IR SpectroscopyEditorial Note: Some figures have been redacted due to copyright issues.

REVIEWER COMMENTS

Reviewer #1 (Remarks to the Author):

The authors propose and experimentally demonstrate wavelength-multiplexed hook nanoantenna array for SEIRAS.

Their work combines detailed theoretical analysis, experiments involving many chips and wide ranges of analytes in both dry and liquid environments.

I think this work is a nice addition to the growing field of nanostructure-based SEIRA. I have some questions and comments for improvements:

1. I appreciate the authors's careful analysis of antenna resonance effects and strategies to optimize SEIRAS performance.

Ultimately, however, the enhancement (3 orders of magnitude) observed from thin-film measurements is not much better than what others have reported using simpler structures (gold nanorod antennas or coupled nanorods). To justify the use of more complex design/fabrication strategies for their hook structures, the authors should provide side-by-side comparison of their results against these previous results and show clear improvements.

2. In the title and main text, the authors highlight 'machine learning enabled mid-IR spectroscopy'. At first this reviewer was intrigued, but in my view the capability demonstrated herein (using relatively simple mixtures) does not go beyond conventional principal component analysis and does not live up to the hype. The authors mentioned that their platform can pave the way to achieve global molecular identification and training with DNN. Until they can show such impressive results, my recommendation is to tone down their claims related to 'machine learning'.

3. Related to Figure 2 FDTD results, the authors's results show that "the enhancement of the electric field of all HNA devices is at the order of 10^5 " and the maximum enhancement of 832891x.

- To avoid confusion, the authors should specify that this is for field intensity enhancement (relating to $|E/E_0|^2$), not the field amplitude.

- I wonder if these field enhancement values may be too optimistic. Could the authors provide more details on how the enhancement value was measured (exactly where in the structure, simulation accuracy etc.). Especially the fact that Fig. 2e(i)(ii)(iii) structures show similar EF is confusing. More details on their FDTD methodology would be helpful.

4. They mention EIT-like, EIA-like, or Fano-like resonance profiles. More detailed and intuitive explanation and relationship to their structure would be helpful.

5. Supporting Information Figure S2

- Instead of just showing color maps, from which is it hard to check detailed field profile, it would be nice to show field amplitude profile at, for example, $Y=0$.

6. Supporting Figure S5 (AFM). It appears that the non-uniformity of antenna thickness is relatively large (65-85 nm), which also shows up in the AFM images. Could the authors mention how such non-uniformity in actual device structure may cause discrepancy between FDTD vs. measured SEIRAS results?

7. Figures are clearly illustrated and well presented, but the text needs thorough revision. Let me point out just a few (mostly minor) issues I could identify:

- abstract: "SEIRAS ... to identify molecular structure" => While SEIRAS can provide absorption bands, they cannot be used to identify molecular structure, strictly speaking.

- abstract is long, yet does not articulate why the performance of WMHNA structures is better than

conventional SEIRAS structures.

- page 3, line 49 "the genetic information" => Did the authors mean 'generic'?
- page 7, line 152 "Lorenz" => Lorentz
- page 11, line 232 "is almost robust among different antenna structures" => What does this mean? almost constant?
- page 13, line 262 " 10^5 " => please specify that this is for the intensity enhancement.
- page 23, line 444 "deep neuron networks (DNN)" => "deep neural network (DNN)"
- page 23, line 450 "the technology gap of SEIRAS in continuous broadband wavelengths" => I don't think the author's work is unique in leveraging nanoantenna supercell structures for broadband SEIRAS.

Reviewer #2 (Remarks to the Author):

Ren and co-workers report on the identification of molecular species using machine learning enabled mid IR spectroscopy. The key result is the identification of mixtures of methanol, ethanol and isopropanol using their technique. The identification seems to be flawless and this is attributed to the effective application of the authors' surface enhanced infrared absorption (SEIRA) spectroscopy technique. The authors introduce a hook-shaped plasmonic nano-particle to create a strong SEIRA effect. By combining multiple hook antennas with slightly different resonances within their device, problems associated with tuning of electromagnetic resonance with molecular absorption lines can somewhat be mitigated. Hook shaped particles of varying dimensions are thus integrated within arrays of super-cells to produce a SEIRA based molecular sensor. The authors call this Wavelength-Multiplexed Hook Nanoantenna Arrays.

The novelty in this work seems to be the use of multiplexed hook antennas to produce a more broadband SEIRA response. The complex spectra that result from different molecules is challenging to interpret so the application of machine learning is a great route to take. This is the subject of much ongoing research, but here its application is timely. The SEIRA technique itself has been well developed for some time. The manuscript does point to routes to overcome some of the challenges of the technique. I believe that the study could be useful to the sensing community. Unfortunately, the manuscript has a few technical problems and the writing is not optimal. The technical problems are raised below. There are also a number of errors and inconsistencies in the manuscript that must be addressed. If the problems can be addressed and with better proof reading, I would be happy to recommend publication in Nature Communications.

I found the introduction quite confusing on first reading as it includes quite a lot of information. On second reading, it makes sense, but I wonder if the author could try to improve the introduction? Currently, it reads a little like a compact literature review with almost 60 references, which is quite a lot for an article of this type. This on its own is not a bad thing, only it makes it difficult to understand what exactly the authors are trying to achieve in this work. At times the motivations for the work are mixed with technical details, which are not directly relevant to the discussion. This makes the introduction a challenging read especially for non-experts. Although I will not insist on a new introduction, a more concise introduction with direct links to the original aspects of the work would aid the readability of the manuscript.

The authors in a number of places refer to an EIT-like response and a Fano-like response. I should point out that all responses are Fano-like. Here the authors refer to the EIT-like case as the one where molecular absorption and antenna resonance are tuned. This is just a special case of the Fano

response. I am pointing this out as it may confuse non-expert readers.

I did not understand the need to distinguish the refractometric effect and the antenna loading effect. The material to be analysed presents a modification of surrounding refractive index, which has a real and imaginary part. The real part of the index modifies the resonant condition of the antenna and so shifts the resonance. The imaginary part induces absorption, which in the case of SEIRA is spectrally localised. The authors refer to a third effect of antenna loading. From Figure 1d iii, the antenna loading effect produces the same shift in antenna resonance as the refractometric effect. The only difference I can see is that the refractometric effect does not exhibit absorption and the antenna loading effect does. But is not the antenna loading effect just the combination of the refractometric and absorption (spectrometric) effects? I thought this was confusing and recommend the authors review their terminology.

In Figure 2, the authors report quite large field enhancement near their hook antenna structures. These enhancements seem to be surprisingly large. The simple bar antenna could be compared to literature where the enhancement should be on the order of a thousand for the wavelength range. (See e.g. Adato et al *Material Today* 18 p 436 (2015).) It looks like the field enhancement might have been accidentally squared? I was also surprised that there is not a stronger variation of the field enhancement for the different hook antenna shapes. I was delighted to see the authors identify the critical coupling point to balance radiative and non-radiative scattering from the antennas with a lovely analysis in Fig 2d - but this does not seem to have any strong effect on the peak field enhancement shown in Fig 2f. The variation here seems to be on the order of a factor of 2 across the range of hook antennas. Can the authors explain this for the benefit of readers?

There are a number of typos and minor errors throughout the manuscript that should be addressed. To sort this out, I suggest a thorough proof-read. Here are some examples:

The term alcoholic molecule is a bit strange – suggest small molecule alcohols.

Line 252 and 629 – bipolar should be dipolar.

Figure 4c. y-label should be Norm. – “Nor.” Is confusing.

Figure 6a. Should y-label be R and not ΔR ?

Line 368 board should be broad

Line 462 Reflectometric should be Refractometric.

Line 471 nuclear acid should be nucleic acid.

Reviewer #3 (Remarks to the Author):

In this work, the authors present an approach for biospectroscopy combining plasmonic nanoantennas, microfluidics and data processing. Specifically, hook-shaped nanoantennas are implemented and the antenna geometry is varied to produce a range of resonance wavelengths, enabling wavelength-multiplexed operation. The antennas are utilized to record the surface-enhanced infrared spectra of several basic chemical compounds such as methanol, ethanol, and isopropanol, and a simple machine learning algorithm is applied to differentiate them.

From a technical standpoint, the manuscript clearly presents the proposed approach, and sufficient numerical simulations and experimental data are presented. However, the work fails to provide significant elements of novelty compared to the current state-of-the art, limiting itself merely to combining some well known concepts from the literature. Therefore, I believe that this manuscript is not suitable for publication in a high-impact multidisciplinary journal such as *Nature Communications* and I recommend rejection.

First and foremost, the proposed approach and methodology are extremely similar to a recent paper by A. John-Herpin et al. “Infrared Metasurface Augmented by Deep Learning for Monitoring Dynamics between All Major Classes of Biomolecules”, cited as Reference 13 in the current

manuscript. In the referenced work, many elements of the current manuscript are already comprehensively demonstrated: plasmonic nanoantennas with different resonance wavelengths, molecular spectroscopy and AI-based analysis.

In fact, the previous work by John-Herpin et al. goes significantly beyond the current manuscript. For example, it investigates a complex biological system consisting of four major classes of biomolecules (proteins, nucleic acids, carbohydrates, lipids) instead of a biologically irrelevant model system of methanol, ethanol, and propanol. Furthermore, the AI-based data analysis based on DNNs in Ref. 13 is much more sophisticated than the algorithms proposed in the current manuscript (principal component analysis and supported vector machines).

Similarly, the proposed hook antenna design seems unnecessarily complicated and does not improve surface-enhanced detection performance compared to established wavelength-multiplexed molecular spectroscopy platforms based on, e.g, Fano resonances. For one of the many examples, see C. Wu, et al., "Fano-resonant asymmetric metamaterials for ultrasensitive spectroscopy and identification of molecular monolayers," *Nat Mater* 11, 69–75 (2012).

In summary, the present manuscript fails to significantly extend the state of the art in its main subject areas (nanophotonics/plasmonics, biological application, AI-based data analysis) and therefore does not meet the requirements for publication in *Nature Communications*.

RESPONSE TO REVIEWERS

Reviewer #1 (Remarks to the Author):

The authors propose and experimentally demonstrate wavelength-multiplexed hook nanoantenna array for SEIRAS.

Their work combines detailed theoretical analysis, experiments involving many chips and wide ranges of analytes in both dry and liquid environments.

I think this work is a nice addition to the growing field of nanostructure-based SEIRA. I have some questions and comments for improvements:

We appreciate the reviewer's positive comments on our work. The point-by-point response is shown below.

Question 1.1

I appreciate the authors's careful analysis of antenna resonance effects and strategies to optimize SEIRAS performance.

Ultimately, however, the enhancement (3 orders of magnitude) observed from thin-film measurements is not much better than what others have reported using simpler structures (gold nanorod antennas or coupled nanorods). To justify the use of more complex design/fabrication strategies for their hook structures, the authors should provide side-by-side comparison of their results against these previous results and show clear improvements.

We thank the reviewer's valuable comment. To make a side-by-side comparison with the state-of-the-art nanoantenna sensors, we made a benchmark Table S1 in supplementary Note S1 to compare the sensing performance with the previous works in recent 5 years. When making the benchmark table, we found that all of the works were designed at different wavelengths for sensing different molecules, which makes it

difficult to show a clear improvement of our designs. Therefore, we design another experiment for thin-film PMMA sensing using our HNA devices and simpler structures (gold nanorod antennas and coupled nanorods). Following the original name in the previous papers, we choose two control designs about the standard Fano-resonant structure (Fano-resonant asymmetric metamaterial, FRAMM)¹ and dipolar resonant structure (Nanorod antenna, NA)². To match antenna resonance with the carbonyl stretching from PMMA at the 5.8 μm wavelength, we apply a scaling factor to all geometric parameters for FRAMM and NA reported in previous papers. From the sensing results, our HNA devices show improvement of sensitivity of 28.2% (reflection mode) and 241% (transmission mode) compared with FRAMM, as well as 83.6% (reflection mode) and 1728% (transmission mode) compared with NA. The sensing results are shown in Fig. S1 and a detailed explanation is added in Supplementary Note S1.

Table S1. The benchmark table of MIR nanoantenna sensors

Sensing unit	Optimization methods	Critical dimension	Field enhancement	Sensing analytes	Wavenumber (cm^{-1})	SEIRA Sensing Signal (%)
Bowtie nanoantenna ³	Nanogap	Sub-3 nm gap	10^7	4-NTP	1515, 1335	0.003% for monolayer
				4-MTP	1594, 1485	0.008% for monolayer
Graphene Nanoribbon ⁴	Field confinement by 2D materials	~3 nm thickness	10^4	Protein	1668, 1532	27% for bilayer
Hybrid Graphene Nanorod ⁵	Field confinement by 2D materials	10 nm gap	5×10^4	Glucose	1470	2% for 100 pmol
Nanorod MA ⁶	Hotspot release, Nanogap	30 nm vertical gap	10^9	H ₂ O, Ethanol, Acetone	3390	0.8364%/pM
Cross MA ⁷	Hotspot release, Nanogap	10 nm vertical gap	8×10^8	ODT	2850-2995	36% for 2.8 nm thickness
				Lipids	2850, 2920	10 mOD for bilayer
Nanorod ⁸	Nanogap, Molecules Enrichment	80 nm gap	10^5	Melittin	1650, 1550	10 mOD for 6×10^{-6} M
				Sucrose	1142	20 mOD for 200 mg/mL
				Nucleotides	1230	20 mOD for 200 mg/ML
Nanograting MA ⁹	Hotspot release,		1600	Proline	1400, 1600	10% for 0.2 $\mu\text{g/ml}$,

	Molecules Enrichment	200 nm vertical gap		D-glucose	1400, 1640	5% for 0.1 $\mu\text{g/ml}$
Nanorod ¹⁰	Molecules Enrichment	~ 100 nm gap	10^5	Benzene(gas)	1480	4×10^{-4} OD for 25 ppb
Cross MA ¹¹	Molecules Enrichment	200 nm vertical gap	2500	CO ₂	2347	0.0358%/ppm
				CH ₄	1305	0.0121%/ppm
Cross MA ¹²	Molecules Enrichment	200 nm vertical gap	4900	PECA	1740, 1200	15% for 10 nm thickness
Nanodisk MA ¹³	Molecules Enrichment	200 nm vertical gap	1000	miR-155	1700	0.42% for 500 pM
WMHNA	Loss engineering	~ 200 nm width	10^5	PMMA	1750, 1470, 1180	16% for 20nm
				Silk	1168, 1532, 1200	50% for 50nm
				Methanol	1540-1300	5.18% for 1%
				Ethanol	1540-1170	3% for 1%
				Isopropanol	1540-1120	4.74% for 1%

*MA: Metamaterial Absorber, consisting of metal-isolator-metal (MIM) structure.

Fig. S1 Sensing performance of different devices. **a** Schematic drawing of hook nanoantenna (HNA) devices and control devices, including Fano-resonant asymmetric metamaterials (FRAMM)¹ and nanorod antenna (NA)⁸. **b** The simulated reflection spectrum of all nanoantenna devices with thin-film analytes of poly(methyl methacrylate) (PMMA). All devices are designed to match with the absorption peak of carbonyl stretching at 5.8 μm wavelength. **c**

The extracted reflection difference (ΔR) of different devices. **d** Simulated transmission spectrum of all nanoantenna devices with thin-film analytes of PMMA. **e** The extracted transmission difference (ΔR) of different devices. **f** Normalized sensitivity of different devices for thin-film sensing. HNA devices perform the best sensitivity in reflection mode (HNA-1) and transmission mode (HNA-2).

Question 1.2

In the title and main text, the authors highlight 'machine learning enabled mid-IR spectroscopy'. At first this reviewer was intrigued, but in my view the capability demonstrated herein (using relatively simple mixtures) does not go beyond conventional principal component analysis and does not live up to the hype. The authors mentioned that their platform can pave the way to achieve global molecular identification and training with DNN. Until they can show such impressive results, my recommendation is to tone down their claims related to 'machine learning'.

We appreciate the valuable comment from the reviewer pointing out the machine learning problem. We have added the deep learning results by a DNN with 2 hidden layers and 20 nodes. The detailed results with 100% accuracy are shown in **Supporting Note S7**. Compared with machine learning using PCA and SVM, the DNN is more automatic for data processing and has greater potentials for a further increased dataset. However, DNN also loses some other information about the physical meaning of the data features from WMHNA since it is treated as a black box.

Question 1.3

3. Related to Figure 2 FDTD results, the authors's results show that "the enhancement of the electric field of all HNA devices is at the order of 10^5 " and the maximum enhancement of 832891x.

- To avoid confusion, the authors should specify that this is for field intensity enhancement (relating to $|E/E_0|^2$), not the field amplitude.

We appreciate the careful review of this issue. We have highlighted the enhancement is calculated from the intensity of the electric field in the related content.

- I wonder if these field enhancement values may be too optimistic. Could the authors provide more details on how the enhancement value was measured (exactly where in the structure, simulation accuracy, etc.). Especially the fact that Fig. 2e(i)(ii)(iii) structures show similar EF is confusing. More details on their FDTD methodology would be helpful.

To solve the concern of the reviewer about the field intensity. The EF is calculated by the intensity enhancement $|E/E_0|^2$ and the value labeled in Fig. 2f is the maximum value of EF. We have revised the y label to avoid misunderstanding. The maximum EF occurs at the tip position of HNA where the detailed profile can be found in Fig. S3b.

According to the details of the FDTD software, the simulation setup of the HNA device is described in **Methods-FDTD Simulation**. The finite-difference time-domain (FDTD) method (Lumerical FDTD) is performed to simulate the far-field spectrum and the nearfield distribution of plasmonic hook nanoantennas. The light source is selected as a plane wave to simulate the incidence of light from free space. The incidence angle and polarization state are adjusted to the desired orientation to excite the dipolar mode plasmonic resonance of HNA. The refractive index of CaF_2 is set at 1.38 at wavelengths ranging from 2 μm to 10 μm . The periodic boundary at the x and y-axis (Fig. 2a) is selected to simulate the effect of the periodic antenna array, and the PML boundary is chosen at the z-axis to transport light into free space. The mesh size is set to be 20 nm at X and Y direction and 10 nm at Z direction, which is 10 times smaller than the smallest dimension of nanoantennas. There is also a trade-off between simulation accuracy and simulation time. When we further decrease the mesh size, the simulation time increases dramatically (more than 10 hours), while the results remain almost the same.

Question 1.4

4. They mention EIT-like, EIA-like, or Fano-like resonance profiles. More detailed and intuitive explanation and relationship to their structure would be helpful.

To avoid confusion, we changed to Fano-like in all parts of the manuscript. The EIT-like and EIA-like line shape can be considered as a special case of Fano-like line shape when the resonance wavelengths are matched between nanoantennas and molecules ($\omega_0 = \omega_m$), following the theoretical expression in Eq. 1, 2.

$$T(\omega) = \left| \frac{S_t}{S_{in}} \right|^2 = \left| \frac{j(\omega - \omega_0) + \gamma_a + \frac{\mu^2}{j(\omega - \omega_m) + \gamma_m}}{j(\omega - \omega_0) + (\gamma_a + \gamma_r) + \frac{\mu^2}{j(\omega - \omega_m) + \gamma_m}} \right|^2 \quad (1)$$

$$R(\omega) = \left| \frac{S_r}{S_{in}} \right|^2 = \left| \frac{\gamma_r}{j(\omega - \omega_0) + (\gamma_a + \gamma_r) + \frac{\mu^2}{j(\omega - \omega_m) + \gamma_m}} \right|^2 \quad (2)$$

Question 1.5

Supporting Information Figure S2

- Instead of just showing color maps, from which is it hard to check detailed field profile, it would be nice to show field amplitude profile at, for example, $Y=0$.

We appreciate the valuable comments about the near-field distribution. We label the field distribution along the long and short arms of the HNA. The revised Fig. S2 and Note S2 are also attached below.

To study the near field enhancement of HNA, we perform the FDTD simulation and monitor the electric field intensity and polarity at resonance wavelengths for different devices. The amplitude of the electric field of different devices is plotted in Fig. S2a using the log scale by the equation $\log(|E/E_0|^2)$. The results show that all of the devices have an enhanced field intensity at the magnitude of 10^5 . The extracted field enhancement ($|E/E_0|^2$) at two arms of HNAs (position labeled in Fig. S2a) are shown in Fig. S2b. The normalized the electric field of each device is characterized by E_x in Fig. S2c, showing a fundamental dipole mode of the resonance and the inverse current induced by the short arm of HNA to tune the radiative loss.

Fig. S3 The near-field distribution of HNA at different ΔL . **a** The enhancement of electric field intensity ($|E/E_0|^2$) of different HNAs. **b** The extracted field enhancement ($|E/E_0|^2$) at two arms of HNAs (position labeled in (a)). **c** The normalized electric field with the component at x-direction of different HNAs. The fundamental dipole mode is clearly shown from the polarity of the electric field (except for the extreme case of $\Delta L=0$ in (IV)).

Question 1.6

Supporting Figure S5 (AFM). It appears that the non-uniformity of antenna thickness is relatively large (65-85 nm), which also shows up in the AFM images. Could the authors mention how such non-uniformity in actual device structure may cause discrepancy between FDTD vs. measured SEIRAS results?

We appreciate the valuable comments from the reviewer. After the double-check of the thickness, we found that the step thickness at CC' line should be 66.4 nm (Fig. S6f). The increase of height in the position region of 0-60 nm is because of the non-uniform topology of the substrate and the redeposition of gold nanoparticles during the lift-off process, which can be seen in the 3D view of the nanoantenna in Fig. S6b. Additionally, we perform the FDTD simulation of the HNA device using the measured geometry from the AFM. As shown in Fig. S6g,h, the simulation results of 70 nm antenna thickness match well with the experiment results in both transmission and reflection modes, while the simulation results of 65 nm antenna thickness show a slightly shorter resonance wavelength. Even though we only deposit ~ 65 nm of gold, the redeposition of gold nanoparticle during liftoff increase the equivalent thickness of the nanoantenna, making it respond more like the 70 nm-thick HNA devices.

Fig. S6 **a** The AFM image of one HNA device. **b** The reconstructed 3D model of HNA from the height measurement in (a). **c** The AFM image of WMHNA with a highlight area of one unit cell. **d-f** The height measurement results of three parts of HNA, showing a range of antenna thickness is 65-70 nm. This is caused by the non-uniformity during the gold deposition. **g,h** Simulation and experiment results of transmission (g) and reflection (h) spectrum of HNA showing in (a). The different antenna thicknesses of 65 nm and 70 nm are simulated to compare with the FTIR measurement.

Question 1.7

Figures are clearly illustrated and well presented, but the text needs thorough revision. Let me point out just a few (mostly minor) issues I could identify:

- abstract: "SEIRAS ... to identify molecular structure" => While SEIRAS can provide absorption bands, they cannot be used to identify molecular structure, strictly speaking.

- abstract is long, yet does not articulate why the performance of WMHNA structures is better than conventional SEIRAS structures.

We appreciate the useful suggestion to revise the abstract in a more accurate and readable way. Considering the 2 comments above, we have revised our abstract to:

Infrared (IR) plasmonic nanoantennas (PNAs) are powerful tools to identify molecules by the IR fingerprint absorption from plasmon-molecules interaction. However, the sensitivity and bandwidth of PNAs are limited by the small overlap between molecules and sensing hotspots and the sharp plasmonic resonance peaks. In addition to field enhancement and molecule enrichment, we propose a novel loss engineering method to design hook nanoantennas (HNAs) by investigating another key parameter of damping rate. With the spectral multiplexing of the HNAs from gradient dimension, the wavelength-multiplexed HNAs (WMHNAs) serve as ultrasensitive vibrational probes in a continuous ultra-broadband region (wavelengths from 6 μm to 9 μm). Leveraging the multi-dimensional features from WMHNA, we develop a machine learning method to extract complementary physical and chemical information from molecules. The proof-of-concept demonstration of molecular recognition from mixed alcohols (methanol, ethanol, and isopropanol) shows 100% identification accuracy from the microfluidic integrated WMHNAs. Our work brings a new degree of freedom to optimize PNAs augmented by machine learning towards small-volume, real-time, label-free molecular recognition from various species in low concentrations for chemical and biological diagnostics.

- page 3, line 49 "the genetic information" => Did the authors mean 'generic'?

We have changed it to 'generic'. The revised sentence become:

Mid-infrared (MIR) fingerprint absorption, reflecting the generic information of molecule structures in chemical bonds and functional groups, provides natural optical probes for molecule identification.

- page 7, line 152 "Lorenz" => Lorentz

We have changed it to 'Lorentz'. The revised sentence becomes:

We use finite-difference time-domain (FDTD) methods to simulate the antenna response and manipulate the optical properties (refractive index, n and absorption coefficient, k) of analytes by the Lorentz model.

- page 11, line 232 "is almost robust among different antenna structures" => What does this mean? almost constant?

We have changed it to 'is almost the same among different antenna structures'. The revised sentence becomes:

The γ_a is related to the ohmic loss of plasmonic material (e.g., Au in this work) and is almost the same among different antenna structures.

- page 13, line 262 " 10^5 " => please specify that this is for the intensity enhancement.

We have highlighted the enhancement is for electric intensity. The revised sentence becomes:

The results show that the enhancement of the electric field intensity of all HNA devices is at the order of 10^5 , and the maximum enhancement of the electric field intensity reaches 832891 times.

- page 23, line 444 "deep neuron networks (DNN)" => "deep neural network (DNN)"

We have revised accordingly.

- page 23, line 450 "the **technology gap** of SEIRAS in continuous broadband wavelengths" => I don't think the author's work is unique in leveraging nanoantenna supercell structures for broadband SEIRAS.

We deleted this expression.

Reviewer #2 (Remarks to the Author):

Ren and co-workers report on the identification of molecular species using machine learning enabled mid IR spectroscopy. The key result is the identification of mixtures of methanol, ethanol and isopropanol using their technique. The identification seems to be flawless and this is attributed to the effective application of the authors' surface enhanced infrared absorption (SEIRA) spectroscopy technique. The authors introduce a hook-shaped plasmonic nano-particle to create a strong SEIRA effect. By combining multiple hook antennas with slightly different resonances within their device, problems associated with tuning of electromagnetic resonance with molecular absorption lines can somewhat be mitigated. Hook shaped particles of varying dimensions are thus integrated within arrays of super-cells to produce a SEIRA based molecular sensor. The authors call this Wavelength-Multiplexed Hook Nanoantenna Arrays.

The novelty in this work seems to be the use of multiplexed hook antennas to produce a more broadband SEIRA response. The complex spectra that result from different molecules is challenging to interpret so the application of machine learning is a great route to take. This is the subject of much ongoing research, but here its application is timely. The SEIRA technique itself has been well developed for some time. The manuscript does point to routes to overcome some of the challenges of the technique. I believe that the study could be useful to the sensing community. Unfortunately, the manuscript has a few technical problems and the writing is not optimal. The technical problems are raised below. There are also a number of errors and inconsistencies in the manuscript that must be addressed. If the problems can be addressed and with better proof reading, I would be happy to recommend publication in Nature Communications.

We appreciate the positive comments from the reviewer. Please check the point-by-point response below.

Question 2.1

I found the introduction quite confusing on first reading as it includes quite a lot of information. On second reading, it makes sense, but I wonder if the author could try to improve the introduction? Currently, it reads a little like a compact literature review with almost 60 references, which is quite a lot for an article of this type. This on its own is not a bad thing, only it makes it difficult to understand what exactly the authors are trying to achieve in this work. At times the motivations for the work are mixed with technical details, which are not directly relevant to the discussion. This makes the introduction a challenging read especially for non-experts. Although I will not insist on a new introduction, a more concise introduction with direct links to the original aspects of the work would aid the readability of the manuscript.

We appreciate the reviewer's comment on the issues of our introduction. We have reorganized the introduction parts to make them more concise and readable. Please check the revised manuscript.

Question 2.2

The authors in a number of places refer to an EIT-like response and a Fano-like response. I should point out that all responses are Fano-like. Here the authors refer to the EIT-like case as the one where molecular absorption and antenna resonance are tuned. This is just a special case of the Fano response. I am pointing this out as it may confuse non-expert readers.

We appreciate the reviewers pointing out this issue. We revised all descriptions to Fano-like to avoid any confusion.

Question 2.3

I did not understand the need to distinguish the refractometric effect and the antenna loading effect. The material to be analysed presents a modification of surrounding refractive index, which has a real and imaginary part. The real part of the index modifies the resonant condition of the antenna and so shifts the resonance. The imaginary part induces absorption, which in the case of SEIRA is spectrally localised. The authors refer to a third effect of antenna loading. From Figure 1d iii, the antenna loading effect produces the same shift in antenna resonance as the refractometric effect. The only difference I can see is that the refractometric effect does not exhibit absorption and the antenna loading effect does. But is not the antenna loading effect just the combination of the refractometric and absorption (spectrometric) effects? I thought this was confusing and recommend the authors review their terminology.

We agree with the reviewer on the physics/optics model of the molecule-antenna coupling system, which are the modification of surrounding refractive index with both real-part and imaginary part. However, the feature of the spectral data after introducing molecule on the antenna is complicated. First of all, the antenna loading effect is not a simple combination of refractometric and absorption effects. It explains the case that the reflection change is different when the resonance of nanoantenna and molecular absorption is detuned. As detuning increases, the molecular absorption changes from SEIRA decrease while the intrinsic absorptivity of the molecule is unchanged. This effect reflects the physical properties of the molecule. This phenomenon is very important for analyzing the mixture of multiple molecules. Taking alcohols

identifications experiment as an example, the water molecules remain the same for different mixtures, but the different reflection changes of water absorption peak also contain the information of the refractive index of liquid samples. This is a different degree-of-freedom of features of the spectrum rather than the wavelength shift of the antenna resonance peak. This feature is also classified as one dimension of the principal component in PCA analysis, while the wavelength shift caused by the refractometric effect and the absorption change caused by the spectroscopic effect are classified as another dimensions of the principal component.

Question 2.4

In Figure 2, the authors report quite large field enhancement near their hook antenna structures. These enhancements seem to be surprisingly large. The simple bar antenna could be compared to literature where the enhancement should be on the order of a thousand for the wavelength range. (See e.g. Adato et al *Material Today* 18 p 436 (2015).) It looks like the field enhancement might have been accidentally squared?

Our field enhancement is calculated using the same methods from the paper mentioned by the review, while the presentation is different. In that particular paper, the authors use the average field enhancement to calculate the mean value of the enhancement in one unit cell of the nanoantenna. In our work together with many other papers^{14,15}, the maximum field enhancement is utilized to characterize the near-field properties for nanoantenna sensing. It carries the same information and does not conflict with each other.

Question 2.5

I was also surprised that there is not a stronger variation of the field enhancement for the different hook antenna shapes. I was delighted to see the authors identify the critical coupling point to balance radiative and non-radiative scattering from the antennas with a lovely analysis in Fig 2d - but this does not seem to have any strong effect on the peak field enhancement shown in Fig 2f. The variation here seems to be on

the order of a factor of 2 across the range of hook antennas. Can the authors explain this for the benefit of readers?

We appreciate the reviewer give us a chance to explain the near-field properties of HNAs. First and foremost, the maximum electric field region is close to the two poles of HNAs according to the dipolar resonance mode. When the folding degree of HNA increase, the distance of two poles decrease following the change of ΔL . The critical point is at the half-fold state where ΔL equals $0.8 \mu\text{m}$. As ΔL change from $1.6 \mu\text{m}$ to $0.8 \mu\text{m}$, the field enhancement does not change much. However, when ΔL changes from $0.8 \mu\text{m}$ to 0 , the strong overlap of the electric field near the two poles of HNA accumulates the field intensity. Therefore, the field has been increased by the order of two.

We also added the related part to the content related to Fig. 2f in the manuscript.

There is a slight increment of the intensity (~ 2 times) from half to full fold of HNAs due to the superposition of the electric field near the two poles of the HNAs. However, compared to the dramatic change of radiative loss, the loss ratio (f) is a more dominant affecting parameter in different HNAs.

Fig. S3 The near-field distribution of HNA at different ΔL . **a** The enhancement of electric field intensity ($|E/E_0|^2$) of different HNAs. **b** The extracted field enhancement ($|E/E_0|^2$) at two arms of HNAs (position labeled in (a)). **c** The

normalized electric field with the component at x-direction of different HNAs. The fundamental dipole mode is clearly shown from the polarity of the electric field (except for the extreme case of $\Delta L=0$ in (IV)).

Question 2.6

There are a number of typos and minor errors throughout the manuscript that should be addressed. To sort this out, I suggest a thorough proof-read. Here are some examples:

The term alcoholic molecule is a bit strange – suggest small molecule alcohols.

We have revised to ‘small molecule alcohols’ or ‘alcohols’ accordingly.

Line 252 and 629 – bipolar should be dipolar.

We have revised it to ‘dipolar’.

Figure 4c. y-label should be Norm. – “Nor.” Is confusing.

We revised all ‘Nor.’ to ‘Norm.’ in Fig. 4c.

Figure 6a. Should y-label be R and not ΔR ?

We revised the y-label in Fig.6a to ‘Reflection (%)’.

Line 368 board should be broad

Line 462 Reflectometric should be Refractometric.

Line 471 nuclear acid should be nucleic acid.

We revised the above typos accordingly. Thanks for the careful check of the reviewer.

Reviewer #3 (Remarks to the Author):

In this work, the authors present an approach for biospectroscopy combining plasmonic nanoantennas, microfluidics and data processing. Specifically, hook-shaped nanoantennas are implemented and the

antenna geometry is varied to produce a range of resonance wavelengths, enabling wavelength-multiplexed operation. The antennas are utilized to record the surface-enhanced infrared spectra of several basic chemical compounds such as methanol, ethanol, and isopropanol, and a simple machine learning algorithm is applied to differentiate them.

Question 3.1

From a technical standpoint, the manuscript clearly presents the proposed approach, and sufficient numerical simulations and experimental data are presented. However, the work fails to provide significant elements of novelty compared to the current state-of-the art, limiting itself merely to combining some well known concepts from the literature. Therefore, I believe that this manuscript is not suitable for publication in a high-impact multidisciplinary journal such as Nature Communications and I recommend rejection.

We appreciate the review's comment about our work. However, there is some misunderstanding of the novelty of work so that we would like to clarify with the reviewer about our achievements.

1. A novel hook nanoantennas (HNAs) structure to improve the sensitivity of enhanced IR absorption fingerprints of molecular vibration for the first time demonstration.
2. A design methodology to optimize the HNAs sensitivity by loss engineering of plasmonic properties based on temporal coupled-mode theory.
3. The proposed WMHNA works at 6 μm to 9 μm wavelengths for continuous ultra-broadband detection of enhanced molecular vibrational spectra for multiple (>15) fingerprint absorption peaks.
4. An ML-based method (PCA and SVM) for extracting multi-dimensional information of complementary physical and chemical properties of molecules from three significant effects – the refractometric effect (RE), spectroscopic effect (SE), and antenna loading effect (ALE).

To the best of our knowledge, none of these four features are reported by the previous papers and they are not well-known as mentioned by the reviewer.

Question 3.2

First and foremost, the proposed approach and methodology are extremely similar to a recent paper by A. John-Herpin et al. “Infrared Metasurface Augmented by Deep Learning for Monitoring Dynamics between All Major Classes of Biomolecules”, cited as Reference 13 in the current manuscript. In the referenced work, many elements of the current manuscript are already comprehensively demonstrated: plasmonic nanoantennas with different resonance wavelengths, molecular spectroscopy and AI-based analysis. In fact, the previous work by John-Herpin et al. goes significantly beyond the current manuscript. For example, it investigates a complex biological system consisting of four major classes of biomolecules (proteins, nucleic acids, carbohydrates, lipids) instead of a biologically irrelevant model system of methanol, ethanol, and propanol. Furthermore, the AI-based data analysis based on DNNs in Ref. 13 is much more sophisticated than the algorithms proposed in the current manuscript (principal component analysis and supported vector machines).

We appreciate the review to provide us with the reference to compare. Indeed, there are many differences between their work and ours, and our work does have improvement of the related technologies. In the paper by A. John-Herpin et al., the author proposes a multi-resonant grating order-coupled nanogap (MR-GONG) design for three resonance wavelengths of 8.33 μm , 6.25 μm , and 3.44 μm . In terms of the device design, it is a combination of commonly used nanorod antennas. According to the antenna loading effect, the maximum enhancement of molecular vibration occurs at the wavelengths near the antenna resonance wavelengths. We also agree that this paper demonstrates a complex biological system consisting of four major classes of biomolecules (proteins, nucleic acids, carbohydrates, lipids). However, it is much easier to distinguish different molecules from different classes because of the large distinction of vibrational fingerprints. In our work, we investigate a novel HNA design with novel design methodology by engineering the radiative loss of nanoantenna. We also multiplex 16 different elements of HNA to form the WMHNA platform covering the MIR fingerprint region from 6 μm to 9 μm wavelengths to minimize the antenna loading effect for broadband detection. For the molecule identification of alcohols, the fingerprint

absorption peaks are strongly overlapped due to the similar chemical structures with the same chemical bonds and functional groups. It is changing and meaningful to identify the molecules in the same class since all of them are significant biomarkers in biological systems. Even though our current demonstration is at the proof-of-concept stage, the WMHNA is a very promising technique to identify more species of molecules in our future studies.

Additionally, we also want to defend the utilization of machine learning. Machine learning or deep learning algorithms serve as analytical tools for SEIRA sensing applications. The figure of merit to judge an algorithm is always the accuracy of identification results rather than the complexity of the algorithm itself. In our results, we achieve 100% identification accuracy using principal component analysis and support vector machines, which the reviewer believes is less sophisticated than the DNN method in Ref. 13. However, we believe this is a strength rather than a weakness by solving the problem using a simple method with 100% accuracy.

Furthermore, we have built up a DNN model to achieve molecule identification from the WMHNA spectra. A detailed explanation is included in the supplementary Note S7.

Our DNN dataset comprises 300×1272 data points and we randomly split these data, in which 80% for training data set, 20% for testing data set. The proposed DNN model is built using the Sequential model of Python's Keras frame. Each fully connected layer followed with ReLu as activation function and the final model was compiled by 'categorical_crossentropy' as the loss function and Adam as the optimizer. We set the batch size as 20. In the 800 epochs, the loss between prediction and ground truth shows excellent convergence in both of training and testing set, and there is no overfitting occurring either. (Fig. S8c) Finally, 100% identification accuracy is obtained for alcohols identification shown in Fig. S8d,e.

Fig. S8 Alcohol identification from microfluidics integrated WMHNA using DNN. **a** The spectral data of different combinations of 1% alcohol solvent. **b** The proposed DNN model with two hidden layers and 20 nodes for identification of 6 different analytes. **c,d** the loss and accuracy of the DNN model at the different number of epochs. **e** The confusion map of final results for alcohol identification.

Compared with machine learning using PCA and SVM, the DNN is more automatic for data processing and has more potentials for a further increased dataset. However, DNN also loses some other information about the physical meaning of the data features from WMHNA since it is treated as a black box. We believe that it is a proper time to judge which method is superior. More experiments need to be investigated in our future work.

Question 3.3

Similarly, the proposed hook antenna design seems unnecessarily complicated and does not improve surface-enhanced detection performance compared to established wavelength-multiplexed molecular spectroscopy platforms based on, e.g. Fano resonances. For one of the many examples, see C. Wu, et al.,

"Fano-resonant asymmetric metamaterials for ultrasensitive spectroscopy and identification of molecular monolayers," *Nat Mater* 11, 69–75 (2012).

We appreciate the concerns raised by the reviewer about the sensitivity of hook nanoantenna. To prove the improvement of sensing performance, we design an experiment of thin-film PMMA sensing using our HNA and the device in the above-mentioned literature, namely Fano-resonant asymmetric metamaterials (FRAMM). Since the FRAMM in the literature is designed for protein sensing at 1400 cm^{-1} ($7.14\text{ }\mu\text{m}$ in wavelength), we match the resonance wavelength of FRAMM with carbonyl stretching from PMMA at the $5.8\text{ }\mu\text{m}$ wavelength by applying a scaling factor (S) to all geometric parameters of FRAMM. The FDTD simulation results of 20nm-thick PMMA thin film are shown in Fig. S1. From the sensing results, our HNA devices show improvement of sensitivity by 28.2% (reflection mode) and 241% (transmission mode) compared with FRAMM. From the obvious improvement above, the HNA device is a good candidate for ultrasensitive MIR molecular sensing. The sensing performance can be optimized from the radiative loss of HNA by tuning the folding degree of the HNA structure (ΔL , defined by L_1-L_3). According to the optimal loss rates at transmission ($f=0.5$) and reflection($f=2$) modes, two HNA devices with different ΔL are designed as optimized structures for transmission (HNA-2) and reflection (HNA-1) modes. The design methodology of HNA shows a comprehensive explanation of the coupling behavior between nanoantenna resonance and molecular vibration, where the loss ratio (f) plays a significant role for spectral sensing signals (spectrum difference). Our loss engineering method opens a new window to optimize the nanoantenna sensing performance by tuning the loss ratio from the structural change, which can be widely used to explore many other antenna geometries rather than HNA. Furthermore, the loss engineer method does not conflict with other optimization methods of field enhancement (nanogap or 2D materials), hotspot release, and molecule enrichment. We will explore the combination of our loss engineering method with other methods to gain the sensing performance of nanoantennas in our future works.

Fig. S1 Sensing performance of different devices. **a** schematic drawing of hook nanoantenna (HNA) devices and control devices, including Fano-resonant asymmetric metamaterials (FRAMM)¹ and nanorod antenna (NA)⁸. **b** The simulated reflection spectrum of all nanoantenna devices with thin-film analytes of poly (methyl methacrylate) (PMMA). All devices are designed to match with the absorption peak of carbonyl stretching at 5.8 μm wavelength. **c** The extracted reflection difference (ΔR) of different devices. **d** Simulated transmission spectrum of all nanoantenna devices with thin-film analytes of PMMA. **e** The extracted transmission difference (ΔT) of different devices. **f** Normalized sensitivity of different devices for thin-film sensing. HNA devices perform the best sensitivity in reflection mode (HNA-1) and transmission mode (HNA-2).

Question 3.4

In summary, the present manuscript fails to significantly extend the state of the art in its main subject areas (nanophotonics/plasmonics, biological application, AI-based data analysis) and therefore does not meet the requirements for publication in Nature Communications.

We appreciate the review's harsh but valuable comments. We have tried our best to improve our manuscript with extra effort and data. We hope the review to reconsider our manuscript with the improved quality compared with our previous submission.

References:

1. Wu, C. *et al.* Fano-resonant asymmetric metamaterials for ultrasensitive spectroscopy and identification of molecular monolayers. *Nat. Mater.* **11**, 69–75 (2012).
2. Adato, R. & Altug, H. In-situ ultra-sensitive infrared absorption spectroscopy of biomolecule interactions in real time with plasmonic nanoantennas. *Nat. Commun.* **4**, 2154 (2013).
3. Dong, L. *et al.* Nanogapped Au Antennas for Ultrasensitive Surface-Enhanced Infrared Absorption Spectroscopy. *Nano Lett.* **17**, 5768–5774 (2017).
4. Rodrigo, D. *et al.* Mid-infrared plasmonic biosensing with graphene. *Science (80-.)*. **349**, 165–168 (2015).
5. Zhu, Y. *et al.* Optical conductivity-based ultrasensitive mid-infrared biosensing on a hybrid metasurface. *Light Sci. Appl.* **7**, 67 (2018).
6. Xu, J. *et al.* Nanometer-Scale Heterogeneous Interfacial Sapphire Wafer Bonding for Enabling Plasmonic-Enhanced Nanofluidic Mid-Infrared Spectroscopy. *ACS Nano* **14**, 12159–12172 (2020).
7. Hwang, I. *et al.* Ultrasensitive Molecule Detection Based on Infrared Metamaterial Absorber with Vertical Nanogap. *Small Methods* **5**, 2100277 (2021).
8. John-Herpin, A., Kavungal, D., Mücke, L. & Altug, H. Infrared Metasurface Augmented by Deep Learning for Monitoring Dynamics between All Major Classes of Biomolecules. *Adv. Mater.* **33**, 2006054 (2021).
9. Miao, X., Yan, L., Wu, Y. & Liu, P. Q. High-sensitivity nanophotonic sensors with passive trapping of analyte molecules in hot spots. *Light Sci. Appl.* **10**, 5 (2021).
10. Nuñez, J., Boersma, A., Grand, J., Mintova, S. & Sciacca, B. Thin Functional Zeolite Layer

- Supported on Infrared Resonant Nano-Antennas for the Detection of Benzene Traces. *Adv. Funct. Mater.* **31**, 2101623 (2021).
11. Zhou, H. *et al.* Metal–Organic Framework-Surface-Enhanced Infrared Absorption Platform Enables Simultaneous On-Chip Sensing of Greenhouse Gases. *Adv. Sci.* **7**, 2001173 (2020).
 12. Li, D. *et al.* Multifunctional Chemical Sensing Platform Based on Dual-Resonant Infrared Plasmonic Perfect Absorber for On-Chip Detection of Poly(ethyl cyanoacrylate). *Adv. Sci.* **2101879**, 2101879 (2021).
 13. Hui, X. *et al.* Infrared Plasmonic Biosensor with Tetrahedral DNA Nanostructure as Carriers for Label-Free and Ultrasensitive Detection of miR-155. *Adv. Sci.* **8**, 2100583 (2021).
 14. Rodrigo, D. *et al.* Resolving molecule-specific information in dynamic lipid membrane processes with multi-resonant infrared metasurfaces. *Nat. Commun.* **9**, 2160 (2018).
 15. John-Herpin, A., Kavungal, D., von Mücke, L. & Altug, H. Infrared Metasurface Augmented by Deep Learning for Monitoring Dynamics between All Major Classes of Biomolecules. *Adv. Mater.* **33**, (2021).

REVIEWER COMMENTS

Reviewer #1 (Remarks to the Author):

The authors have addressed all of my comments and I recommend this paper for publication in Nature Communications.

Reviewer #3 (Remarks to the Author):

In their resubmission, the authors have improved multiple technical aspects of their manuscript. However, there are still significant concerns regarding the novelty and the performance of the proposed method, an aspect that was also raised in question 1.1 of reviewer #1.

Below, I reproduce the arguments of the authors regarding novelty together with my comments:

Author argument 1. A novel hook nanoantennas (HNAs) structure to improve the sensitivity of enhanced IR absorption fingerprints of molecular vibration for the first time demonstration.

Whereas the geometrical shape of the hook has not appeared in the literature before, it is still not clear how the authors' design improves upon the best available current designs. The newly added discussion around Fig. S1 shows only a marginal improvement in performance compared to the established FRAMM design from 2012 in reflection geometry. Experimentally, there is no conceptual difference between measurements in reflection and transmission, so only the best performance should be compared. Furthermore, it is surprising to me that the comparison system "nanorod" in Fig. S1 is a comparatively old design (Adato, R. & Altug, H. Nat. Commun. 4, 2154 (2013)) and no comparison is made with recent optimized nanoantenna systems for SEIRA such as in the John-Herpin et al. paper mentioned in my original review. Such recent concepts greatly increase the sensitivity by engineering the nanogaps between the antennas and their radiative coupling. Therefore, the current performance comparison in Fig. S1 is not fair and misrepresents the authors' performance claims. Furthermore, recent SEIRA approaches based on high-performance Coaxial Zero-Mode Resonators are completely neglected in the comparison (see, e.g., D. Yoo et al. High-Contrast Infrared Absorption Spectroscopy via Mass-Produced Coaxial Zero-Mode Resonators with Sub-10 nm Gaps. Nano Lett 18, 1930–1936 (2018)).

Author argument 2. A design methodology to optimize the HNAs sensitivity by loss engineering of plasmonic properties based on temporal coupled-mode theory.

SEIRA absorption engineering via tailoring of the losses (critical coupling) has been conclusively shown in R. Adato et al., Engineered Absorption Enhancement and Induced Transparency in Coupled Molecular and Plasmonic Resonator Systems, Nano Lett. 2013, 13, 6, 2584–2591 (2013). Likewise, description of SEIRA effects using TCMT is ubiquitous in the literature (see, e.g., recent review papers such as F. Neubrech et al., Surface-Enhanced Infrared Spectroscopy Using Resonant Nanoantennas. Chem Rev 117, 5110–5145 (2017)).

Author argument 3. The proposed WMHNA works at 6 μm to 9 μm wavelengths for continuous ultra-broadband detection of enhanced molecular vibrational spectra for multiple (>15) fingerprint absorption peaks.

As the authors themselves observe in their simulations in Fig. S1, the previous plasmonic designs can easily be modified to cover a spectral range from 6 μm to 9 μm through straightforward geometrical scaling. So broadband operation is not an argument for novelty.

Author argument 4. An ML-based method (PCA and SVM) for extracting multi-dimensional

information of complementary physical and chemical properties of molecules from three significant effects – the refractometric effect (RE), spectroscopic effect (SE), and antenna loading effect (ALE).

The simultaneous analysis of RE, SE, and ALE is interesting, but does not significantly extend the state-of-the-art in terms of detection limit or sensing functionality.

In summary, I maintain my assessment regarding the novelty of the authors' work as well as the conceptual similarity to the previous literature and recommend rejection of the manuscript.

We would like to thank for reviewer's time to provide the comments and made the revisions which we hope can meet your requirement for publication. In general, we have done extra simulations and experiments to collect new data and added the additional discussion into the main manuscript and supplementary in response to reviewers' comments. We have added **Figure S7 in Note S6, revised Table S1 in Note S1**. All of the **revisions** in the main manuscript and the supplementary information are marked in **red**, and the **point-by-point responses** to the reviewers' comments are provided in the following pages of this letter.

Reviewer #3 (Remarks to the Author):

In their resubmission, the authors have improved multiple technical aspects of their manuscript. However, there are still significant concerns regarding the novelty and the performance of the proposed method, an aspect that was also raised in question 1.1 of reviewer #1.

Below, I reproduce the arguments of the authors regarding novelty together with my comments: **“Author argument 1. A novel hook nanoantennas (HNAs) structure to improve the sensitivity of enhanced IR absorption fingerprints of molecular vibration for the first time demonstration.”**

Comment 1.1

Whereas the geometrical shape of the hook has not appeared in the literature before, it is still not clear how the authors' design improves upon the best available current designs. The newly added discussion around Fig. S1 shows only a marginal improvement in performance compared to the established FRAMM design from 2012 in reflection geometry. Experimentally, there is no conceptual difference between measurements in reflection and transmission, so only the best performance should be compared.

Reply:

The mechanisms and methods to design hook nanoantenna are shown in **Fig. 2 and Fig. 3** with detailed theoretical modeling (Page 10 Line 189 to Page 13 Line 260 in **Manuscript** and Note S2 in **Supplementary Information**) and experimental proof (Page 14 Line 261 to Page 15 Line 293 in **Manuscript** and Note S4-S6 in **Supplementary Information**). Different from the previous paper, we propose a new methodology to optimize the sensing performance by engineering loss ratio f using hook nanoantenna, which does not conflict with available current designs manipulating the coupling strength μ .¹⁻⁵ The study of sensing improvement is based on controlled variables where the μ remains almost fixed among different hook designs, not on the best available designs. In terms of transmission and reflection, we agree that there is no conceptual difference between the two modes. However, based on our methodology, the different rules occur when designing the plasmonic sensor in different modes, which are $f=0.5$ in transmission mode and $f=2$ in reflection mode. Our work investigates the theoretical model for this phenomenon and provides design methodology for future work to look at antenna loss when designing the plasmonic sensors.

Comment 1.2

Furthermore, it is surprising to me that the comparison system “nanorod” in Fig. S1 is a comparatively old design (Adato, R. & Altug, H. Nat. Commun. 4, 2154 (2013)) and no comparison is made with

recent optimized nanoantenna systems for SEIRA such as in the John-Herpin et al. paper mentioned in my original review. Such recent concepts greatly increase the sensitivity by engineering the nanogaps between the antennas and their radiative coupling. Therefore, the current performance The comparison in Fig. S1 is not fair and misrepresents the authors' performance claims.

Reply:

We thank the reviewer's comment and provided the information related to the paper of John-Herpin et al. in Table S1 in the R1 supplementary. To provide a fair comparison with the nanoantennas reported in John-Herpin et al.⁵ paper, we have further made new experiments to collect real testing data in order to compare the sensing performance. The related discussion is added in Note S6 Fig. S7.

Fig. S7 a,b PMMA sensing testing results of grating order nanogap (GONG) designs and HNA designs in reflection (a) and transmission mode (b). c the extracted normalized sensitivity of different sensors. HNA-1 shows an improvement of 50% in reflection mode and HNA-2 shows an improvement of 200% in transmission mode. By engineering the loss of HNA from geometric design, the sensing performance can be improved without the introduction of nanogap. d-f SEM image of fabricated GONG sensor with the nanogap of 60 nm(d), HNA-1 (e), and HNA-2 (f) sensors.

In brief, the nanorod is the typical dipole nanoantenna and the resonance wavelength is determined by the length of the nanorod. However, the optimization process of the nanorod sensor at the designed wavelength is to increase coupling strength μ by reducing the gap between adjacent nanorods to nanogap (normally <100 nm). It means a smaller nanogap is desired in order to enhance μ . However, such design and optimization rules make the nanoantenna device fabrication become challenged eventually. Besides, the fabrication cost increases dramatically because of using high-resolution lithography tools. The tradeoff between fabrication and sensing performance needs to be considered for plasmonic nanoantenna sensors. To overcome this grand challenge, we develop a new design method of loss engineering to avoid the need of using nanogap to enhance μ . We work on another parameter loss rate f . More importantly, our key contribution of introducing hook nanoantennas is to investigate the basic phenomenon behind this parameter f and to propose a methodology that can manipulate this parameter f to optimize sensing performance. **The whole optimization process is achieved by simply changing the geometry of nanoantennas without introducing special materials (e.g. enrichment layer⁶, 2D materials⁷, etc.) or special processes to make super-narrow nanogap (<100 nm).**

In summary, we have clearly presented the detailed design methodology and its difference from the previous works in our introduction (Page 3, Line 51 to Line 66 and Page 5, Line 101 to Line 107) and discussion (“Design Principles of Hook Nanoantenna” Page 10, Line 189 to Page 13, Line 260 including Fig.2). With the comparison with the state-of-the-art nanogap design from the paper of John-Herpin et. al.⁵, our HNA sensors show the improved sensing performance in the top-rank of many research works, which give a general guideline for plasmonic nanoantenna sensor design towards general better results without complicated processes for nanogap.

Comment 1.3

Furthermore, recent SEIRA approaches based on high-performance Coaxial Zero-Mode Resonators are completely neglected in the comparison (see, e.g., D. Yoo et al. High-Contrast Infrared Absorption Spectroscopy via Mass-Produced Coaxial Zero-Mode Resonators with Sub-10 nm Gaps. Nano Lett 18, 1930–1936 (2018)).

Reply:

We thank the review’s comments on the paper by D. Yoo et. al.⁴ This paper is cited by Ref. 24 in our manuscript from first submission. The author D. Yoo et. al. propose a great work on SEIRA enhancement made by sub-10 nm gap from mass-produced coaxial zero-mode resonators and demonstrate good performance on silk protein sensing. However, there is some limitation of this paper. As mentioned in our reply in Comment 1.2, the nanogap formation is challenging due to the limitation of lithography. Therefore, a complicated process is proposed in the paper as shown in (Fig Redacted) reproduced from the paper. Finally, a quasi-3D plasmonic structure with vertical nanogap is formed by special ALD, ion milling, and sacrificial etching process. Additionally, since the nanogap is only 7nm, macromolecules like protein antibody, exosome, virus, etc. are not accessible to sensing hotspots inside the nanogap, limiting the application for biomolecular diagnostics. This paper is added to benchmark table S1 for sensing performance comparison. However, due to the special fabrication shown in (Fig Redacted) compared with our process in (Fig Redacted), it is difficult for us to make the device for experimental comparison. Even if this paper shows better performance than our HNAs for thin-film sensing, our sensors also show great capabilities for liquid-based biosensing without the above-mentioned drawbacks. Not to mention that our work also has the advances of ultra-broadband detection range and machine learning enhanced analysis.

In summary, our design only required one-lithography step in its fabrication process. Moreover, we do not need to create a sub-100nm nanogap to achieve good sensing performance. Therefore, our platform is not only scalable in large array sensor devices in terms of uniformity and repeatability but more cost-effective as well.

(Fig Redacted) The fabrication process flow of Coaxial Zero-Mode Resonators.

(Fig Redacted) The fabrication process flow of our plasmonic nanoantenna sensors. The Au pattern is subject to change according to the layout drawing of HNA or WMHNA.

Author argument 2. A design methodology to optimize the HNAs sensitivity by loss engineering of plasmonic properties based on temporal coupled-mode theory.

Comment 2

SEIRA absorption engineering via tailoring of the losses (critical coupling) has been conclusively shown in R. Adato et al., Engineered Absorption Enhancement and Induced Transparency in Coupled Molecular and Plasmonic Resonator Systems, *Nano Lett.* 2013, 13, 6, 2584–2591 (2013). Likewise, description of SEIRA effects using TCMT is ubiquitous in the literature (see, e.g., recent review papers such as F. Neubrech et al., Surface-Enhanced Infrared Spectroscopy Using Resonant Nanoantennas, *Chem Rev* 117, 5110–5145 (2017)).

Reply:

We agree that using TCMT to analyze SEIRA has already been reported by previous papers, where we have cited these papers in our manuscript since the first submitted version. However, the key point is that the relationship between loss rate f and sensing performance is not well developed and investigated so far. As we discussed in **Fig. 2a,b**, there are three approaches to optimize nanoantenna sensing in previous works by increasing coupling efficiency μ . **Hotspots release** is proposed to increase accessible sensing areas for analytes, which are usually blocked by the substrate, to further enhance the coupling efficiency. Thanks to the ultra-confined electric field of PNAs, the coupling region only covers hundreds of nanometers near the antenna surface. Therefore, the **molecule enrichment** method is utilized to accumulate localized molecules in the effective sensing area. Another approach to enhance the nearfield coupling is to increase the electric field intensity by squeezing the adjacent nanoantenna into the **nanogap**. In this case, the narrower the nanogap is, the stronger the PNAs enhancement behave. However, all of these three methods introduce extra processes or materials to make the plasmonic sensors, which increase the fabrication cost or limit the working condition. Therefore, we investigate the relationship between loss and sensing performance and propose a new methodology to design plasmonic sensors based on f , which is totally different from the paper mentioned by the reviewer.⁸

Fig.2 Design principle of hook nanoantenna. **a** The derived intensity changes in transmission and reflection mode with the perfect match of molecular absorption and antenna resonance by TCMT. **b** The summaries of methods affecting the coupling strength between molecules and antenna by previous works. **c** The proposed hook antenna for sensing optimization by tuning the radiative loss and the effect of loss ratio to intensity change in transmission and reflection mode.

To be specific, the paper by R. Adato et al.⁸ only study the effect of sensing line shape change from the under-coupled region ($f < 1$) to the over-coupled region ($f > 1$) for two-port system of metal-insulator-metal (MIM) structures as shown in (**Fig Redacted**) below.

(Fig Redacted) **a** The MIM structure used in their study; **b** loss of nanoantenna for different gaps; **c** sensing performance for different gaps.

We also quote the description in this paper “Given that the structures are otherwise virtually identical, this effect (lineshape change shown in **(Fig Redacted)**) is unexpected from the standard intuition behind, for example, SEIRA spectroscopy, which predicts absorption signal to scale with the near-field enhancement generated by a plasmonic resonance. In future applications, leveraging plasmonic enhancement and coupling between bright and dark modes, it is thus evident that exactly what quantity is in fact being measured must be considered carefully and the effects predicted by the description here be taken into account.”

From this description, the author only studies that loss rate f , which is manipulated by the gap of MIM absorber in this work, can affect the sensing performance in addition to a near-field coupling for MIM structures, resulting in the lineshape change from EIT to EIA. They also encourage the following work to consider this effect to the sensor design but they do not provide the quantitative relationship between loss rate f and sensing performance to tell people how to design the sensor based on this effect.

However, in single layer nanoantenna structure, there is no similar line-shape change as described in the paper of R. Adato et al.⁸ in transmission and reflection mode. Therefore, the effect mentioned by the authors by changing the vertical gaps does not exist in the single-layer nanoantenna structures in our work. Alternatively, we use the hook nanoantenna to tailor the loss rate f by changing ΔL . Furthermore, we study the effect of loss rate f on sensing performance and further derived the optimal sensing condition of plasmonic nanoantenna in transmission ($f=0.5$) and reflection ($f=2$) modes for single layer nanoantenna structures in **Fig. 2a-c**, which can be considered as other parameters to optimize the sensing performance in addition to tailoring the coupling efficiency μ .

In addition to the contribution of loss-engineered HNA, we also leverage the HNA structure in broadband WMHNA designs to achieve continuous detection from 6 μm to 9 μm . Furthermore, we developed a machine learning method for simultaneous analysis of RE, SE, and ALE.

Author argument 3. The proposed WMHNA works at 6 μm to 9 μm wavelengths for continuous ultra-broadband detection of enhanced molecular vibrational spectra for multiple (>15) fingerprint absorption peaks.

Comment 3

As the authors themselves observe in their simulations in Fig. S1, the previous plasmonic designs can easily be modified to cover a spectral range from 6 μm to 9 μm through straightforward geometrical scaling. So broadband operation is not an argument for novelty.

Reply:

We thank the reviewer’s comment. First of all, our broadband design enabled by wavelength-multiplexed hook nanoantenna (WMHNA) provide a methodology of generic design for universal sensing application. Our work is an engineering breakthrough to combine 16 different nanoantenna structures with gradient geometries covering continuous broadband from 6 μm to 9 μm , which can be further extended to a broader range by adding more nanoantenna structures or increasing the gradient. Furthermore, the gradient structure is helpful to compensate for the antenna loading effect and all of the working regions can provide a great improvement of the SEIRA signal since at least one of the elements works at the molecular vibration wavelengths. We have proven the claim from thin-film sensing and

liquid dynamic monitoring in the section in manuscript “Wavelength Multiplexed Hook Nanoantenna Array” and “Liquid Dynamics Monitoring with Broadband Fingerprint Absorption” (Page 15, Line 294 to Page 19, Line 372)

(Fig Redacted) Two typical examples of broadband design by different nanoantenna elements. **a** Dual-band nanorod design work at $\sim 3\text{-}3.5\ \mu\text{m}$ (Peak 2) & $\sim 5.5\text{-}6.5\ \mu\text{m}$ (Peak 1). **b** Multi-resonant grating-order coupled nanogap design for triple-band operation at $\sim 3\text{-}3.5\ \mu\text{m}$ (Peak 3), $\sim 5.5\text{-}6.5\ \mu\text{m}$ (Peak 2) & $\sim 7.5\text{-}9\ \mu\text{m}$ (Peak 1).

As shown in (Fig Redacted), the current broadband design of nanoantenna sensors is achieved by separating resonance peaks from different nanoantenna elements. The separation of resonance peaks limits the application of sensors to very limited absorption of molecules and needs to be redesigned if the target analytes change. For example, the dual-band nanorod design (Fig Redacted) works at $\sim 3\text{-}3.5\ \mu\text{m}$ (Peak 2) & $\sim 5.5\text{-}6.5\ \mu\text{m}$ (Peak 1) by engineering two different lengths of nanorod designs. This dual-band design is dedicated to the detection of Amide I, II, and CH_2 , CH_3 .⁹

Multi-resonant grating-order coupled nanogap (MR-GONG) design (Fig Redacted) work for triple-band operation at $\sim 3\text{-}3.5\ \mu\text{m}$ (Peak 3), $\sim 5.5\text{-}6.5\ \mu\text{m}$ (Peak 2) & $\sim 7.5\text{-}9\ \mu\text{m}$ (Peak 1).⁵ The MR-GONG consists of a dual-band GONG operating at Peak 1 and Peak 3 and a single band GONG working at Peak 2, which means one operating wavelength is determined by one length selection of the GONG sensor. The antenna loading effect (ALE) is dominated to limit the sensing performance by only having three different designs in this broadband range. In the $3.5\ \mu\text{m}\text{-}5.5\ \mu\text{m}$ (between Peak 3 & Peak 2) and $6.5\ \mu\text{m}\text{-}7.5\ \mu\text{m}$ (between Peak 2 & Peak 1) wavelength range, MR-GONG may also collect some absorption signals but the performance is poor due to the ALE illustrated in Fig. 4a-c.

Fig.4 Experiment characterization of WMHNA to decrease loading effect with broadband absorption enhancement. **a** The testing reflection spectra of hook nanoantenna with PMMA thin film at different L. **b** The baseline-corrected reflection spectra from (a). **c** The normalized sensitivity of hook nanoantenna devices with different L indicates the loading effect. Sensitivity becomes maximum when the two wavelengths of molecular vibration and antenna resonance are matched.

Fig.6 a The broadband spectra of WMHNA under different analytes. All alcohol solvents are diluted to 1% in DI water. b The extracted absorption spectra of different alcohol solvents from WMHNA. c The corresponding second-order derivative of absorption spectra in (b).

However, the IR fingerprint absorption is continuously distributed in the spectrum. Therefore, we leverage 16 designs of HNA with different lengths to make a supercell of WMHNA. Our WMHNA operating at continuous broadband from 6-9 μm makes a meaningful milestone for the global sensing platform since many of the molecules contain similar chemical structures and perform overlapped absorption peaks in certain wavelengths. As demonstrated in Fig. 6a-c, the WMHNA can capture more than 15 different fingerprint absorption peaks in one measurement for alcohol identification, which is difficult to achieve by the dual/triple-band design shown in (Fig Redacted). Moreover, our WMHNA can be easily adapted to generic molecular detection without changing the design, while the design shown in (Fig Redacted) needs to redesign the operating wavelengths by revising the length of nanoantennas to tune for different molecular sensing applications.

Last but not least, the elements to form the broadband devices are hook nanoantenna, which is optimized for sensing by loss engineering and proved to have better performance in Fig. S7 than the MR-GONG shown in (Fig Redacted). Therefore, the WMHNA also performs great sensing results in the broadband region, which is a perfect platform to further conduct machine learning for molecular identification.

Author argument 4. An ML-based method (PCA and SVM) for extracting multi-dimensional information of complementary physical and chemical properties of molecules from three significant effects – the refractometric effect (RE), spectroscopic effect (SE), and antenna loading effect (ALE).

Comment 4

The simultaneous analysis of RE, SE, and ALE is interesting, but does not significantly extend the state-of-the-art in terms of detection limit or sensing functionality.

Reply:

We appreciate the reviewer's comment but may not agree with this claim. As mentioned in our reply to Comment 3, the WMHNA is a platform for universal molecular detection, especially for multiple

components analysis thanks to the extraordinary molecular selectivity. Simultaneous analysis of RE, SE, and ALE can further enhance the molecular selectivity from SEIRA spectrum, which is meaningful and helpful when the analyte becomes complex and data volume becomes large. In the real application, the molecular species and concentration of analytes may be unknown. Therefore, it is difficult to make the traditional peak analysis of specific wavelengths to identify the molecules. A complex analysis considering different features from RE, SE, and ALE definitely helps to improve the actual accuracy in the applications. In this case, machine learning is a great tool to extract the data features from different dimensions in a fast and accurate way. Although there are a few works using machine learning to analyze the SEIRA signal, we are the first ones to consider different features from RE, SE, and ALE simultaneously. One example is from the paper of John-Herpin et. al.⁵ shown in **Fig. 2-5a,b**. Before sending SEIRA data to DNN, they extract the feature of SE by removing the RE from ALS fitting. The related description is quoted below:

“SEIRAS spectra typically appear on a skewed baseline (Figure 3a, (Fig Redacted)) as a consequence of spectral shifting of the antenna resonance wavelengths throughout the experiment, which is due to the effective refractive index change in the antenna vicinity as analytes are introduced. We take this effect into account in the preprocessing of the raw spectra where we subtract an asymmetric least squares (ALS) fit to have the real-time absorbance signals centered around zero (Figure 3b, (Fig Redacted)).”

The other paper of L. Kühner et. al.¹⁰ demonstrates a machine learning method by analyzing the RE and SE, as shown in **Fig. 2-5c**. The related description is quoted below:

“First and second principal component as obtained from PCA of our measurement cycle. The first component represents the spectral shift of the plasmonic resonance due to different effective refractive indices, whereas the second component comprises the vibrational information as also highlighted by the color-coded bars.”

(Fig Redacted) Examples of adoption of machine learning/ deep learning in SEIRA analysis. **a, b** The SEIRA data processing before deep neural network (DNN) before **(a)** and after **(b)** ALS fitting to extract the SE features by removing the effect from RE and ALE. **c** The PCA analysis of SEIRA data to extract features from RE and SE.

Although the theoretical detection limit or sensing functionality is determined by the performance of plasmonic nanoantenna sensors and testing systems (e.g. light sources, photodetectors, etc.), the simultaneous analysis of RE, SE, and ALE bring in the different features of SEIRA sensors in addition to the pure analysis of absorption value or line shape, which is meaningful and helpful when the analyte becomes complex and data volume become large.

In summary, I maintain my assessment regarding the novelty of the authors' work as well as the conceptual similarity to the previous literature and recommend rejection of the manuscript.

Reply:

Table R2-1 Comparison of previous papers with our work

Paper	Optimization	Operation wavelengths	Field enhancement	Sensing analytes	Machine/Deep Learning	Remarks
Nano Lett. 13, 6, 2584–2591 (2013) ⁸	Loss engineering (f)	5.7 μm	10^4	PMMA thin film	No	No quantitative study of sensing performance as a function of loss ratio (f)
Nano Lett. 18, 1930–1936 (2018) ⁴	Nanogap (μ), 7nm gap	6-6.5 μm	10^7	Silk	No	The difficult process to make nanogap, limited area for sensing hotspot
Nature Commun. 9, 2160 (2018) ⁹	Enrichment (μ)	~3-3.5 μm & ~5.5-6.5 μm	10^6	Lipid member, Melittin, Streptavidin, GABA	No	Limited bandwidth, only detect amide I, II and CH_2 , CH_3 absorption
ACS Sensors 4, 1973–1979 (2019) ¹⁰	N.A.	~9.2-9.8 μm	N.A.	Glucose, Fructose	Yes, machine learning analysis of SE, RE.	Limited sensing performance and bandwidth
Advanced Materials 33, 2006054 (2021) ⁵	Enrichment & Nanogap (μ), 80 nm gap	~3-3.5 μm , ~5.5-6.5 μm & ~7.5-9 μm	10^5	Lipids, Melittin, Sucrose, Nucleotides	Yes, deep learning analysis of SE.	Slightly limited bandwidth for universal sensing in the fingerprint region, Only considering SE in deep learning.
Our work	Loss engineering (f)	Continuous 6-9 μm	10^5	PMMA, Silk, Acetone, Methanol, Ethanol, Isopropanol	Yes, machine learning and deep learning of SE, RE, and ALE simultaneously	A new design methodology based on loss engineering, Ultrabroadband continuous detection for the global platform, complex analysis of features from SE, RE, ALE using machine learning and deep learning

In summary, our work shows great advances and contributions in three aspects.

Firstly, we decode the complex information measured in SEIRA with machine learning and deep learning and, for the first time, reveal the different roles of refractometric effect, spectroscopic effect, and antenna loading effect (**Novel point 1**). The complex SEIRA spectra can be partially understood with the aid of our theoretical framework of loss engineering, which further guides our design of hook nanoantennas (**Novel point 2**). Our method of continuous broadband design is validated even when our device is extremely miniaturized, where the nanoantennas array is no more periodic, which again spells

out the power of our method in dealing with complex SEIRA spectra especially for multi-component molecular identification (**Novel point 3**).

We appreciate the reviewers' time and comments. We hope the reviewer reconsider our manuscript in our revision.

References:

1. Dong, L. *et al.* Nanogapped Au Antennas for Ultrasensitive Surface-Enhanced Infrared Absorption Spectroscopy. *Nano Lett.* **17**, 5768–5774 (2017).
2. Hwang, I. *et al.* Ultrasensitive Molecule Detection Based on Infrared Metamaterial Absorber with Vertical Nanogap. *Small Methods* **2100277**, 2100277 (2021).
3. Huck, C. *et al.* Surface-Enhanced Infrared Spectroscopy Using Nanometer-Sized Gaps. *ACS Nano* **8**, 4908–4914 (2014).
4. Yoo, D. *et al.* High-Contrast Infrared Absorption Spectroscopy via Mass-Produced Coaxial Zero-Mode Resonators with Sub-10 nm Gaps. *Nano Lett.* **18**, 1930–1936 (2018).
5. John-Herpin, A., Kavungal, D., Mücke, L. & Altug, H. Infrared Metasurface Augmented by Deep Learning for Monitoring Dynamics between All Major Classes of Biomolecules. *Adv. Mater.* **33**, 2006054 (2021).
6. Zhou, H. *et al.* Metal–Organic Framework-Surface-Enhanced Infrared Absorption Platform Enables Simultaneous On-Chip Sensing of Greenhouse Gases. *Adv. Sci.* **2001173**, 1–11 (2020).
7. Rodrigo, D. *et al.* Mid-infrared plasmonic biosensing with graphene. *Science.* **349**, 165–168 (2015).
8. Adato, R., Artar, A., Erramilli, S. & Altug, H. Engineered Absorption Enhancement and Induced Transparency in Coupled Molecular and Plasmonic Resonator Systems. *Nano Lett.* **13**, 2584–2591 (2013).
9. Rodrigo, D. *et al.* Resolving molecule-specific information in dynamic lipid membrane processes with multi-resonant infrared metasurfaces. *Nat. Commun.* **9**, 2160 (2018).
10. Kühner, L. *et al.* Vibrational Sensing Using Infrared Nanoantennas: Toward the Noninvasive Quantitation of Physiological Levels of Glucose and Fructose. *ACS Sensors* **4**, 1973–1979 (2019).

REVIEWERS' COMMENTS

Reviewer #3 (Remarks to the Author):

I am satisfied with the revisions and the manuscript can be published as is.